# ESTIMATING RIEMANNIAN METRIC WITH NOISE-CONTAMINATED INTRINSIC DISTANCE

## ABSTRACT

We extend metric learning by studying the Riemannian manifold structure of the underlying data space induced by dissimilarity measures between data points. The key quantity of interest here is the Riemannian metric, which characterizes the Riemannian geometry and defines straight lines and derivatives on the manifold. Being able to estimate the Riemannian metric allows us to gain insights into the underlying manifold and compute geometric features such as the geodesic curves. We model the observed dissimilarity measures as noisy responses generated from a function of the intrinsic geodesic distance between data points. A new local regression approach is proposed to learn the Riemannian metric tensor and its derivatives based on a Taylor expansion for the squared geodesic distances. Our framework is general and accommodates different types of responses, whether they are continuous, binary, or comparative, extending the existing works which consider a single type of response at a time. We develop theoretical foundation for our method by deriving the rates of convergence for the asymptotic bias and variance of the estimated metric tensor. The proposed method is shown to be versatile in simulation studies and real data applications involving taxi trip time in New York City and MNIST digits.

## 1 INTRODUCTION

The estimation of distance metric, also known as metric learning, has attracted great interest since its introduction for classification (Hastie & Tibshirani, 1996) and clustering (Xing et al., 2002). A global Mahalanobis distance is commonly used to obtain the best distance for discriminating two classes (Xing et al., 2002; Weinberger & Saul, 2009). While a global metric is often the focus of earlier works, multiple local metrics (Frome et al., 2007; Weinberger & Saul, 2009; Ramanan & Baker, 2011; Chen et al., 2019) are found to be useful because they better capture the data space geometry. There is a great body of work on distance metric learning; see, e.g., Bellet et al. (2015); Suárez et al. (2021) for recent reviews.

Metric learning is intimately connected with learning on Riemannian manifolds. Hauberg et al. (2012) connects multi-metric learning to learning the geometric structure of a Riemannian manifold, and advocates its benefits in regression and dimensional reduction tasks. Lebanon (2002; 2006); Le & Cuturi (2015) discuss Riemannian metric learning by utilizing a parametric family of metric, and demonstrate applications in text and image classification. Like in these works, our target is to learn the *Riemannian metric* instead of the *distance metric*, which fundamentally differentiates our approach from most existing works in metric learning. We focus on the nonparametric estimation of the data geometry as quantified by the Riemannian metric tensor. Contrary to distance metric learning, where the coefficient matrix for the Mahalanobis distance is constant in a neighborhood, the Riemannian metric is a smooth tensor field that allows analysis of finer structures. Our emphasis is in inference, namely learning how differences in the response measure are explained by specific differences in the predictor coordinates, rather than obtaining a metric optimal for a supervised learning task.

A related field is manifold learning which attempts to find low-dimensional nonlinear representations of apparent high-dimensional data sampled from an underlying manifold (Roweis & Saul, 2000; Tenenbaum et al., 2000; Coifman & Lafon, 2006). Those embedding methods generally start by assuming the local geometry is given, e.g., by the Euclidean distance between ambient data

points. Not all existing methods are isometric, so the geometry obtained this way can be distorted. Perraul-Joncas & Meila (2013) uses the Laplacian operator to obtain pushforward metric for the low-dimensional representations. Instead of specifying the geometry of the ambient space, our focus is to learn the geometry from noisy measures of intrinsic distances. Fefferman et al. (2020) discusses an abstract setting of this task, while our work proposes a practical estimation of the Riemannian metric tensor when coordinates are also available, and we show that our approach is numerically sound.

We suppose that data are generated from an unknown Riemannian manifold, and we have available the coordinates of the data objects. The Euclidean distance between the coordinates may not reflect the underlying geometry. Instead, we assume that we further observe similarity measures between objects, modeled as noise-contaminated intrinsic distances, that are used to characterize the intrinsic geometry on the Riemannian manifold. The targeted Riemannian metric is estimated in a data-driven fashion, which enables estimating geodesics (straight lines and locally shortest paths) and performing calculus on the manifold.

To formulate the problem, let $(\mathcal{M}, G)$ be a Riemannian manifold with Riemannian metric $G$, and $dist(\cdot, \cdot)$ be the geodesic distance induced by $G$ which measures the true intrinsic difference between points. The coordinates of data points $x_0, x_1 \in \mathcal{M}$ are assumed known, identifying each point via a tuple of real numbers. Also observed are noisy measurements $y$ of the intrinsic distance between data points, which we refer to as *similarity measurements* (equivalently dissimilarity). The response is modeled flexibly, and we consider the following common scenarios: (i) noisy distance, where $y = dist(x_0, x_1)^2 + \epsilon$ for error $\epsilon$, (ii) similarity/dissimilarity, where $y = 0$ if the two points $x_0, x_1$ are considered similar and $y = 1$ otherwise, and (iii) relative comparison, where a triplet of points $(x_0, x_1, x_2)$ are given and $y = 1$ if $x_0$ is more similar to $x_1$ than to $x_2$ and $y = 0$ otherwise. The binary similarity measurement is common in computer vision (e.g. Chopra et al., 2005), while the relative comparison could be useful for perceptual tasks and recommendation system (e.g. Schultz & Joachims, 2003; Berenzweig et al., 2004). We aim to estimate the Riemannian metric $G$ and its derivatives using the coordinates and similarity measures among the data points.

The major contribution of this paper is threefold. First, we formulate a framework for probabilistic modeling of similarity measurements among data on manifold via intrinsic distances. Based on a Taylor expansion for the spread of geodesic curves in differential geometry, the local regression procedure successfully estimates the Riemannian metric and its derivatives. Second, a theoretical foundation is developed for the proposed method including asymptotic consistency. Last and most importantly, the proposed method provides a geometric interpretation for the structure of the data space induced by the similarity measurements, as demonstrated in the numerical examples that include a taxi travel and an MNIST digit application.

## 2 BACKGROUND IN RIEMANNIAN GEOMETRY

For brevity, *metric* now refers to *Riemannian metric* while *distance metric* is always spelled out. Throughout the paper, $\mathcal{M}$ denotes a $d$-dimensional manifold endowed with a coordinate chart $(U, \varphi)$, where $\varphi : U \rightarrow \mathbb{R}^d$ maps a point $p \in U \subset \mathcal{M}$ on the manifold to its coordinate $\varphi(p) = (\varphi^1(p), \ldots, \varphi^d(p)) \in \mathbb{R}^d$. Without loss of generality, we identify a point by its coordinate as $(p^1, \ldots, p^d)$, suppressing $\varphi$ for the coordinate chart. Upper-script Roman letters denote the components of a coordinate, e.g., $p^i$ is the $i$-th entry in the coordinate of the point $p$, and $\gamma^i$ is the $i$-th component function of a curve $\gamma : \mathbb{R} \supset [a, b] \rightarrow \mathcal{M}$ when expressed on chart $U$. The tangent space $T_p\mathcal{M}$ is a vector space consisting of velocities of the form $v = \gamma'(0)$ where $\gamma$ is any curve satisfying $\gamma(0) = p$. The coordinate chart induces a basis on the tangent space $T_p\mathcal{M}$, as $\partial_i|_p = \partial/\partial x^i|_p$ for $i = 1, \ldots, d$, so that a tangent vector $v \in T_p\mathcal{M}$ is represented as $v = \sum_{i=1}^d v^i \partial_i$ for some $v^i \in \mathbb{R}$, suppressing the subscript $p$ in the basis. We adopt the Einstein summation convention unless otherwise specified, namely $v^i \partial_i$ denotes $\sum_{i=1}^d v^i \partial_i$, where common pairs of upper- and lower-indices denotes a summation from 1 to $d$ (see e.g., Lee, 2013, pp.18–19).

The Riemannian metric $G$ on a $d$-dimensional manifold $\mathcal{M}$ is a smooth tensor field acting on the tangent vectors. At any $p \in \mathcal{M}$, $G(p) : T_p\mathcal{M} \times T_p\mathcal{M} \rightarrow \mathbb{R}$ is a symmetric bi-linear tensor/function satisfying $G(p)(v, v) \geq 0$ for any $v \in T_p\mathcal{M}$ and $G(p)(v, v) = 0$ if and only if $v = 0$. On a chart $\varphi$, the metric is represented as a $d$-by-$d$ positive definite matrix that quantifies the distance traveled

along infinitesimal changes in the coordinates. With an abuse of notation, the chart representation of $G$ is given by the matrix-valued function $p \mapsto G(p) = [G_{ij}(p)]_{i,j=1}^d \in \mathbb{R}^{d \times d}$ for $p \in \mathcal{M}$, so the distance traveled by $\gamma$ at time $t$ for a duration of $dt$ is $[G_{ij}(\gamma(t))\dot{\gamma}^i(t)\dot{\gamma}^j(t)]^{1/2}$. The intrinsic distance induced by $G$, or the *geodesic distance*, is computed as

$$dist\,(p,q) = \inf_\alpha \int_0^1 \sqrt{\sum_{1 \leq i,j \leq d} G_{ij}(\alpha(t))\dot{\alpha}^i(t)\dot{\alpha}^j(t)}dt, \qquad (2.1)$$

for two points $p, q$ on the manifold $\mathcal{M}$, where infimum is taken over any curve $\alpha : [0,1] \to \mathcal{M}$ connecting $p$ to $q$.

A *geodesic curve* (or simply *geodesic*) is a smooth curve $\gamma : \mathbb{R} \supset [a,b] \to \mathcal{M}$ satisfying the *geodesic equations*, represented on a coordinate chart as

$$\ddot{\gamma}^k(t) + \dot{\gamma}^i(t)\dot{\gamma}^j(t)\Gamma_{ij}^k \circ \gamma(t) = 0, \text{ for } i,j,k = 1,\ldots,d, \qquad (2.2)$$

where over-dots represent derivative w.r.t. $t$; $\Gamma_{ij}^k = \frac{1}{2}G^{kl}\left(\partial_i G_{jl} + \partial_j G_{il} - \partial_l G_{ij}\right)$ are the *Christoffel symbols* at $p$; and $G^{kl}$ is the $(k,l)$-element of $G^{-1}$. Solving (2.2) with initial conditions[1] produces geodesic that traces out the generalization of a straight line on the manifold, preserving travel direction with no acceleration, and is also locally the shortest path.

Considering the shortest path $\gamma$ connecting $p$ to $q$ and applying Taylor's expansion at $t = 0$, we obtain $dist\,(p,q)^2 \approx \sum_{1 \leq i,j \leq d} G_{ij}(p)(q^i - p^i)(q^j - p^j)$, showing the connection between the geodesic distance and a quadratic form analogous to the Mahalanobis distance. Our estimation method is based on this approximation, and we will discuss the higher-order terms shortly which unveil finer structure of the manifold.

# 3 LOCAL REGRESSION FOR SIMILARITY MEASUREMENTS

## 3.1 PROBABILISTIC MODELING FOR SIMILARITY MEASUREMENTS

Suppose that we observe $N$ independent triplets $(Y_u, X_{u0}, X_{u1})$, $u = 1, \ldots, N$. Here, the $X_{uj}$ are locations on the manifold identified with their coordinates $\left(X_{uj}^1, \ldots, X_{uj}^d\right) \in \mathbb{R}^d$, $j = 1, 2$, and $Y_u$ are noisy similarity measures of the proximity of $(X_{u0}, X_{u1})$ in terms of the intrinsic geodesic distance $dist\,(\cdot,\cdot)$ on $\mathcal{M}$. To account for different structures of the similarity measurements, we model the response in a fashion analogous to generalized linear models. For $X_{u0}, X_{u1}$ lying in a small neighborhood $\mathcal{U}_p \subset \mathcal{M}$ of a target location $p \in \mathcal{M}$, the similarity measure $Y_u$ is modeled as

$$\mathbb{E}\,(Y_u|X_{u0}, X_{u1}) = g^{-1}\left(dist\,(X_{u0}, X_{u1})^2\right), \qquad (3.1)$$

where $g$ is a given link function that relates the conditional expectation to the squared distance.

*Example* 3.1. We describe below three common scenarios modeled by the general framework (3.1).

1. Continuous response being the squared geodesic distance contaminated with noise:

$$Y_u = dist\,(X_{u0}, X_{u1})^2 + \sigma(p)\varepsilon_n, \qquad (3.2)$$

where $\varepsilon_1, \ldots, \varepsilon_n$ are i.i.d. mean zero random variables, and $\sigma : \mathcal{M} \to \mathbb{R}^+$ is a smooth positive function determining the magnitude of noise near the target point $p$. This model will be applied to model trip time as noisy measure of cost to travel between locations.

2. Binary (dis)similarity response:

$$\mathbb{P}\,(Y_u = 1|X_{u0}, X_{u1}) = \text{logit}^{-1}\left(dist\,(X_{u0}, X_{u1})^2 - \hbar\,(p)\right) \qquad (3.3)$$

for some smooth function $\hbar : \mathcal{M} \to \mathbb{R}$, where $\text{logit}(\mu) = \log\left(\mu/\left(1 - \mu\right)\right), \mu \in (0,1)$ is the logit function. This models the case when there are latent labels for $X_{uj}$ (e.g., digit 6 v.s. 9) and $Y_u$ is a measure of whether their labels are in common or not. The function $\hbar\,(p)$ in (3.3) describes the homogeneity of the latent labels for points in a small neighborhood of $p$. The latent labels could have intrinsic variation even if measurements are made for the same data points $x = X_{u0} = X_{u1}$, and the strength of which is captured by $\hbar\,(p)$.

---

[1]See Appendix B for details about solving it in practice.

3. Binary relative comparison response, where we extend our model for triplets of points $(X_{u0}, X_{u1}, X_{u2})$, where $Y_u$ stands for whether $X_{u0}$ is more similar to $X_{u1}$ than to $X_{u2}$:

$$\mathbb{P}\left(Y_u = 1 | X_{u0}, X_{u1}, X_{u2}\right) = \text{logit}^{-1}\left(dist\left(X_{u0}, X_{u2}\right)^2 - dist\left(X_{u0}, X_{u1}\right)^2\right), \quad (3.4)$$

so that the relative comparison $Y_u$ reflects the comparison of squared distances.

## 3.2 Local Approximation of Squared Distances

We develop a local approximation for the squared distance as the key tool to estimate our model (3.1) through local regression. Proposition 3.1 provides a Taylor expansion for the squared geodesic distance between two geodesics with same starting point but different initial velocities (see Figure D.1 for visualization). For a point $p$ on the Riemannian manifold $\mathcal{M}$, let $\exp_p : T_p\mathcal{M} \to \mathcal{M}$ denote the exponential map defined by $\exp_p(tv) = \gamma(t)$ where $\gamma$ is a geodesic starting from $p$ at time 0 with initial velocity $\gamma'(0) = v \in T_p\mathcal{M}$. For notational simplicity, we suppress the dependency on $p$ in geometric quantities (e.g., the metric $G$ is understood to be evaluated at $p$). For $i = 1, \ldots, d$, denote $\delta^i = \delta^i(t) = \gamma^i(t) - \gamma^i(0)$ as the difference in coordinate after a travel of time $t$ along $\gamma$.

*Proposition* 3.1 (spread of geodesics, coordinated). Let $p \in \mathcal{M}$ and $v, w \in T_p\mathcal{M}$ be two tangent vectors at $p$. On a local coordinate chart, the squared geodesic distance between two geodesics $\gamma_0(t) = \exp_p(tv)$ and $\gamma_1(t) = \exp_p(tw)$ satisfies, as $t \to 0$,

$$dist\left(\gamma_0(t), \gamma_1(t)\right)^2 = \delta^i_{0-1}\delta^j_{0-1}G_{ij} + \delta^i_{0-1}\left(\delta^k_0\delta^l_0 - \delta^k_1\delta^l_1\right)\Gamma^j_{kl}G_{ij} + O(t^4) \quad (3.5)$$

where for $i, j, k, l, m = 1, \ldots, d$,

- $\delta^i_0 = \gamma^i_0(t) - p^i$, $\delta^i_1 = \gamma^i_1(t) - p^i$, and $\delta^i_{0-1} = \delta^i_0 - \delta^i_1$, i.e., $\delta^i_0$, $\delta^i_1$ are differences in $i$-th coordinates of $\gamma_0(t)$ and $\gamma_1(t)$ to the origin $p$, respectively, and $\delta^i_{0-1} = \delta^i_0 - \delta^i_1$ is the coordinate difference between $\gamma_0(t)$ and $\gamma_1(t)$;

- $G_{ij}$ and $\Gamma^j_{kl}$ are the elements of the metric and Christoffel symbols at $p$, respectively.

To the RHS of (3.5), the first term is the quadratic term in distance metric learning. The second term is the result of coordinate representation of geodesics. It vanishes under the *normal coordinate* where the Christoffel symbols are zero[2]. It inspires the use of local regression to estimate the metric tensor and the Christoffel symbols. For $X_{u0}, X_{u1}$ in a small neighborhood of $p$, write the linear predictor as

$$\eta_u := \beta^{(0)} + \delta^i_{u,0-1}\delta^j_{u,0-1}\beta^{(1)}_{ij} + \delta^k_{u,0-1}\left(\delta^i_{u0}\delta^j_{u0} - \delta^i_{u1}\delta^j_{u1}\right)\beta^{(2)}_{ijk}, \quad (3.6)$$

a function of the intercept $\beta^{(0)}$ and coefficients $\beta^{(1)}_{ij}, \beta^{(2)}_{ijk}$, where $\delta^i_{u0} = X^i_{u0} - p^i$, $\delta^i_{u1} = X^i_{u1} - p^i$, and $\delta^i_{u,0-1} = \delta^i_{u0} - \delta^i_{u1}$, for $i, j, k, l = 1, \ldots, d$, and $u = 1, \ldots, N$. The intercept term $\beta^{(0)}$ is included for capturing $\hbar(p)$ in (3.3) under that scenario and can otherwise be dropped from the model. The link function connects the linear predictor to the conditional mean via $\mu_u := g^{-1}(\eta_u) \approx \mathbb{E}(Y_u | X_{u0}, X_{u1})$ as indicated by (3.1) and (3.5), where $\mu_u$ is seen as a function of the coefficients $\beta^{(0)}, \beta^{(1)}_{ij}$, and $\beta^{(2)}_{ijk}$. Therefore, upon the specification of a loss function $Q : \mathbb{R} \times \mathbb{R} \to \{0\} \cup \mathbb{R}^+$ and non-negative weights $w_1, \ldots, w_N$, the minimizers

$$(\hat{\beta}^{(0)}, \hat{\beta}^{(1)}_{ij}, \hat{\beta}^{(2)}_{ijk}) = \underset{\beta^{(0)}, \beta^{(1)}_{ij}, \beta^{(2)}_{ijk}; i,j,k}{\arg\min} \sum_{u=1}^{N} Q\left(Y_u, \mu_u\right)w_u, \quad (3.7)$$

subject to $\beta^{(1)}_{ij} = \beta^{(1)}_{ji}, \quad \beta^{(2)}_{ijk} = \beta^{(2)}_{jik}$, for $i, j, k, l = 1, \ldots, d,$ \quad (3.8)

are used to estimate the metric tensor and Christoffel symbols, obtaining

$$\hat{G}_{ij} = \hat{\beta}^{(1)}_{ij}, \quad \hat{\Gamma}^l_{ij} = \hat{\beta}^{(2)}_{ijk}\hat{G}^{kl}, \quad (3.9)$$

---

[2]See e.g., Lee (2018) pp. 131–133 for normal coordinate. See Meyer (1989) and Proposition D.1 for coordinate-invariant version of Proposition 3.1.

where $\hat{G}^{kl}$ is the matrix inverse of $\hat{G}$ satisfying $\hat{G}^{kl}\hat{G}_{kj} = 1_{\{j=l\}}$. The symmetry constraints (3.8) are the result of the symmetries in the metric tensor and Christoffel symbols, and are enforced by optimizing over only the lower triangular indices $1 \leq i < j \leq d$ without constraints. Asymptotic results guarantees the positive-definiteness of the metric estimate, as will be shown in Proposition 4.1. To weigh the pairs of endpoints according to their proximity to the target location $p$, we apply kernel weights specified by

$$w_u = h^{-2d} \prod_{i=1}^{d} K\left(\frac{X_{u0}^i - p^i}{h}\right) K\left(\frac{X_{u1}^i - p^i}{h}\right) \tag{3.10}$$

for some $h > 0$ and non-negative kernel function $K$. The bandwidth $h$ controls the bias–variance trade-off of the estimated Riemannian metric tensor and its derivatives.

Altering the link function $g$ and the loss function $Q$ in (3.7) enables flexible local regression estimation for models in Example 3.1.

*Example* 3.2. Consider the following loss functions for estimating the metric tensors and the Christoffel symbols when data are drawn from model (3.2)–(3.4), respectively.

1. Continuous noisy response: use squared loss $Q(y, \mu) = (y - \mu)^2$ with $g$ being the identity link function so $\mu_u = \eta_u$.

2. Binary (dis)similarity response: use log-likelihood of Bernoulli random variable

$$Q(y, \mu) = y \log \mu + (1 - y) \log(1 - \mu), \tag{3.11}$$

and $g$ the logit link, so $\mu_u = \text{logit}^{-1}(\eta_u)$. The model becomes a local logistic regression.

3. Binary relative comparison response: apply the same loss function (3.11) and logit link as in the previous scenario, but here we formulate the linear predictor based on $dist(X_{u0}, X_{u2})^2 - dist(X_{u0}, X_{u1})^2 \approx \eta_{u1} - \eta_{u2}$ and

$$\mu_u = g^{-1}(\eta_{u1} - \eta_{u2}). \tag{3.12}$$

Locally, the difference in squared distances is approximated by

$$\eta_{u1} - \eta_{u2} = \left(\delta_{u,0-1}^i \delta_{u,0-1}^j - \delta_{u,0-2}^i \delta_{u,0-2}^j\right) \beta_{ij}^{(1)} \tag{3.13}$$
$$+ \left(\delta_{u,0-1}^k \left(\delta_{u0}^i \delta_{u0}^j - \delta_{u1}^i \delta_{u1}^j\right) - \delta_{u,0-2}^k \left(\delta_{u0}^i \delta_{u0}^j - \delta_{u2}^i \delta_{u2}^j\right)\right) \beta_{ijk}^{(2)},$$

for $\delta_{u2}^i = X_{u2}^i - p^i$ and $\delta_{u,0-2}^i = \delta_{u2}^i - \delta_{u0}^i$, $i = 1, \ldots, d$. Here $\eta_{u1}$ and $\eta_{u2}$ are constructed in analogy to (3.6) using $(X_{u0}, X_{u1})$ and $(X_{u0}, X_{u2})$ pair respectively.

Examples in Section 5 will further illustrate the proposed method in those scenarios. Besides the models listed, other choices for the link $g$ and loss function $Q$ can also be considered under this local regression framework (Fan & Gijbels, 1996), accommodating a wide variety of data. To efficiently estimate the metric on the entire manifold $\mathcal{M}$, we apply a procedure based on discretization and post-smoothing, as detailed in Appendix B.

## 4 BIAS AND VARIANCE OF THE ESTIMATED METRIC TENSOR

This subsection provides asymptotic justification for model (3.2) with $\mathbb{E}(Y_u | X_{u0}, X_{u1}) = dist(X_{u0}, X_{u1})^2$ under the squared loss $Q(\mu, y) = (\mu - y)^2$ and the identity link $g(\mu) = \mu$. The estimator we analyzed here fits a local quadratic regression without intercept and approximates the squared distance by a simplified form of (3.6):

$$dist(X_{u0}, X_{u1})^2 \approx \eta_u := \delta_{u,0-1}^i \delta_{u,0-1}^j \beta_{ij}^{(1)}, \tag{4.1}$$

for $u = 1, \ldots, N$. Given a suitable order of the indices $i, j$ for vectorization, we rewrite the formulation into a matrix form. Denote the local design matrix and regression coefficients as

$$\mathbf{D}_u = \left(\delta_{u,0-1}^1 \delta_{u,0-1}^1, \ldots, \delta_{u,0-1}^i \delta_{u,0-1}^j, \ldots, \delta_{u,0-1}^d \delta_{u,0-1}^d\right)^T, \quad \boldsymbol{\beta} = \left(\beta_{11}^{(1)}, \ldots, \beta_{ij}^{(1)}, \ldots, \beta_{dd}^{(1)}\right)^T,$$

so that the linear predictor $\eta_u = \mathbf{D}_u^T \boldsymbol{\beta}$. Further, write $\mathbf{D} = (\mathbf{D}_1, \ldots, \mathbf{D}_N)^T$, $\mathbf{Y} = (Y_1, \ldots, Y_N)^T$, and $\mathbf{W} = \text{diag}(w_1, \ldots, w_N)$, with weights $w_u$ specified in (3.10). The objective function in (3.7) becomes $(\mathbf{Y} - \mathbf{D}\boldsymbol{\beta})^T \mathbf{W} (\mathbf{Y} - \mathbf{D}\boldsymbol{\beta})$, and the minimizer is $\hat{\boldsymbol{\beta}} = (\mathbf{D}^T \mathbf{W} \mathbf{D})^{-1} \mathbf{D}^T \mathbf{W} \mathbf{Y}$, for which we will analyze the bias and variance.

To characterize the asymptotic bias and variance of the estimator, we assume the following conditions are satisfied in a neighborhood of the target $p$. These conditions are standard and analogous to those assumed in a local regression setting (Fan & Gijbels, 1996).

(A1) The joint density of endpoints $(X_{u0}, X_{u1})$ is positive and continuously differentiable.

(A2) The functions $G_{ij}, \Gamma_{ij}^k$ are $C^2$-smooth for $i, j, k = 1, \ldots, d$.

(A3) The kernel $K$ in weights (3.10) is symmetric, continuous, and has a bounded support.

(A4) $\sup_u \text{var}(Y_u | X_{u0}, X_{u1}) < \infty$.

*Proposition* 4.1. Under (A1)–(A4), $\text{bias}(\hat{\boldsymbol{\beta}} | \mathbf{X}) = O_p(h^2)$, $\text{var}(\hat{\boldsymbol{\beta}} | \mathbf{X}) = O_p(N^{-1} h^{-4-2d})$, as $h \to 0$ and $Nh^{2+2d} \to \infty$, where $\mathbf{X} = \{(X_{u0}, X_{u1})\}_{u=1}^N$ is the collection of observed endpoints.

The local approximation (4.1) is similar to a local polynomial estimation of the second derivative of a $2d$-variate squared geodesic distance function, explaining the order of $h$ in the rates for bias and variance.

## 5 SIMULATION

We illustrate the proposed method using simulated data with different types of responses as described in Example 3.1. We study whether the proposed method well estimates Riemannian geometric quantities, including the metric tensor, geodesics, and Christoffel symbols. Additional details are included in Appendix C of the Supplementary Materials.

### 5.1 UNIT SPHERE

The usual arc-length/great circle distance on the $d$-dimensional unit sphere is induced by the round metric, which is expressed under the stereographic projection coordinate $(x^1, \ldots, x^d)$ as $\mathring{G}_{ij} = 4 (1 + \sum_{k=1}^d x^k x^k)^{-2} 1_{\{i=j\}}$, for $i, j = 1, \ldots, d$. Under the additive model (3.2) in Example 3.1, we considered either noiseless or noisy responses by setting $\sigma(p) = 0$ or $\sigma(p) > 0$ respectively.

Experiments were preformed with $d = 2$ and the finding are summarized in Figure 5.1. For continuous responses, the left panel of Figure 5.1a visualizes the true and estimated metric tensors via cost ellipses (A.1) and the right panel shows the corresponding geodesics by solving the geodesic equations (2.2) with true and estimated Christoffel symbols. The metrics and the geodesics were well estimated under the continuous response model without or without additive errors, where the estimates overlap with the truth. Figure 5.1b evaluates the relative estimation errors $\|\hat{G} - G\|_F / \|G\|_F$ and $\|\hat{\Gamma} - \Gamma\|_F / \|\Gamma\|_F$ w.r.t. the Frobenius norm (A.2) for data from the continuous model (3.2).

For binary responses under model (3.3), Figure 5.1c visualizes the data where the background color illustrates $\hbar$. Figure 5.1d and left panel of Figure 5.1a suggest that the intercept and the metric were reasonably estimated, while the geodesics are slightly away from the truth (Figure 5.1a, right). This indicates that the binary model has higher complexity and less information is being provided by the binary response (see also Figure C.1b in the Supplementary Materials).

### 5.2 RELATIVE COMPARISON ON THE DOUBLE SPIRALS

A set of $7 \times 10^4$ points on $\mathbb{R}^2$ were generated around two spirals, corresponding to two latent classes $\mathcal{A}$ and $\mathcal{B}$ (e.g., green points in Figure 5.2a are from latent class $\mathcal{A}$). We compare neighboring points $(X_{u0}, X_{u1}, X_{u2})$ to generate relative comparison response $Y_u$ as follows. For $u = 1, \ldots, N$, $Y_u = 1$ if $X_{u0}, X_{u1}$ belong to the same latent class and $X_{u0}, X_{u2}$ belong to different classes;

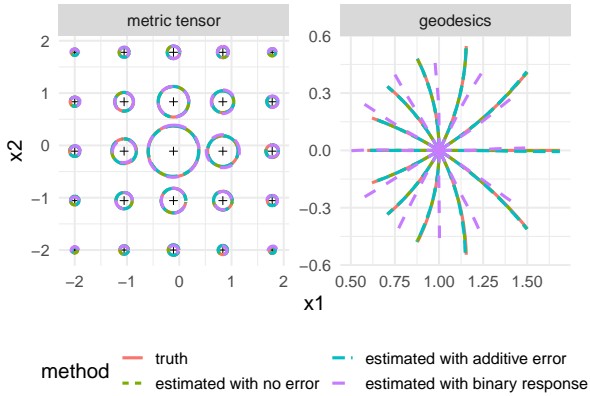

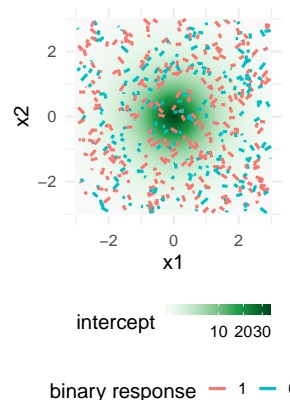

(a) Estimated and true metric tensors using ellipses representation (left) and the geodesic curves (right) starting from $(1, 0)$ with unit initial velocities pointing to 1–12 o'clock directions.

(c) Simulated data, where line segments show pairs of endpoints colored according to their binary responses. The background shows the value of $\hbar$.

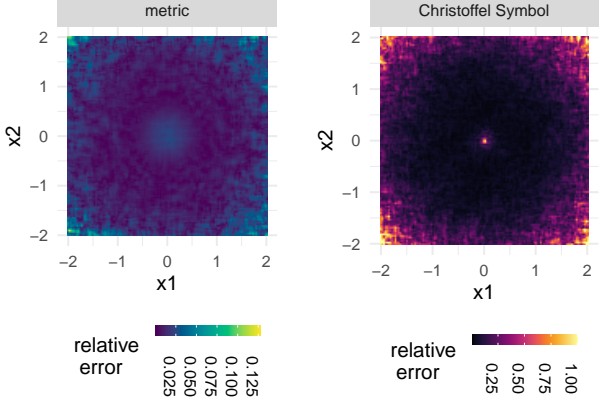

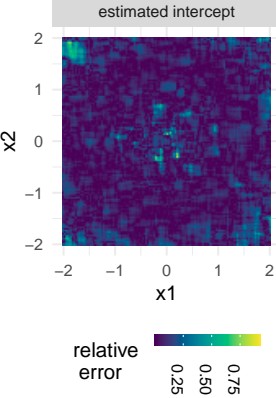

(b) Relative errors in term of Frobenius norm of the estimated tensors for the continuous response model (3.2) with additive error.

(d) Errors for estimating $\hbar$ with binary responses (3.3).

Figure 5.1: Simulation results for 2-dimensional sphere under stereographic projection coordinate.

otherwise $Y = 0$. Figure 5.2b shows a portion of the $N = 6,965,312$ comparison generated, where the hollow circles in the middle of each wedge correspond to $X_{u0}$.

Here, contrast of the two latent classes induces the intrinsic distance, so the distance is larger across the supports of the two classes and smaller within a single support. Therefore, the resulting metric tensor should reflect less cost while moving along the tangential direction of the spirals compared to perpendicular directions. Estimates were drawn under model (3.4) by minimizing the objective (3.7) with the link function (3.12) and the local approximation (3.13).

The estimated metric shown in Figure 5.2c is consistent with the interpretation of the intrinsic distance and metric induced by the class membership discussed above. Meanwhile, the estimated geodesic curve unveils the hidden circular structure of the data support as shown in Figure 5.2d.

## 6 NEW YORK CITY TAXI TRIP DURATION

We study the geometry induced by taxi travel time in New York City (NYC) during weekday morning rush hours. New York City Taxi and Limousine Commission (TLC) Trip Record Data was

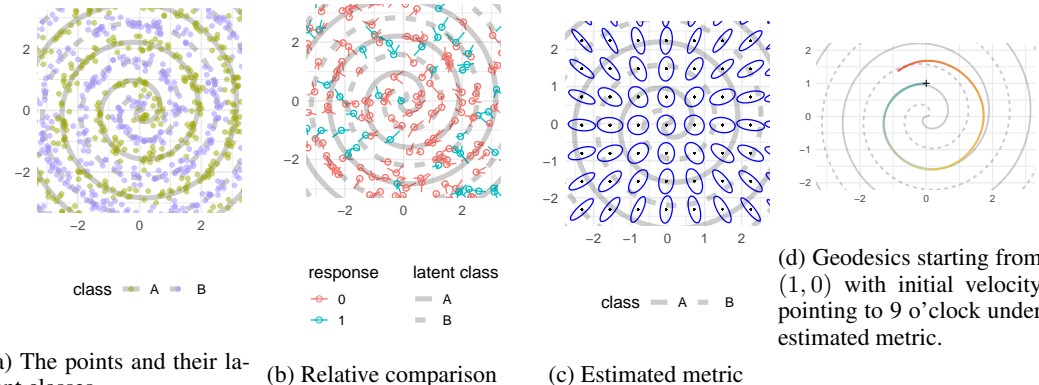

(a) The points and their latent classes

(b) Relative comparison

(c) Estimated metric

(d) Geodesics starting from $(1, 0)$ with initial velocity pointing to 9 o'clock under estimated metric.

Figure 5.2: Simulation results for relative comparison on double spirals. Gray curves (solid and dashed) in the background represent the approximate support of the two latent classes. In (b), the tiny circles are $X_{u0}$, each with two segments connecting to $X_{u1}$ and $X_{u2}$, colored according to $Y_u$.

accessed on May 1st, 2022[3] to obtained business day morning taxi trip records including GPS coordinates for pickup/dropoff locations as $(X_{u0}, X_{u1})$ and trip duration as $Y_u$. Estimation to the travel time metric was drawn under model (3.2) with $Q(y, \mu) = (y - \mu)^2$ and $g(\mu) = \mu$.

Figure 6.1a shows the estimated metric for taxi travel time. The background color shows the Frobenius norm of the metric tensor, where larger values mean that longer travel time is required to pass through that location. Trips through midtown Manhattan and the financial district were estimated to be the most costly during rush hours, which is coherent to the fact that these are the city's primary business districts. Moreover, the cost ellipses represent the cost in time to travel a unit distance along different directions. This suggests that in Manhattan, it takes longer to drive along the east–west direction (narrower streets) compared to the north–south direction (wider avenues).

Geodesic curves in Figure 6.1b show where a 15-minutes taxi ride leads to starting from the Empire State Building. Each geodesic curve corresponds to one of 12 starting directions (1–12 o'clock). Note that we apply a continuous Riemannian manifold approximation to the city, so the geodesic curves provide approximations to the shortest paths between locations and need not conform to the road network. Travel appears to be faster in lower Manhattan than in midtown Manhattan. The spread of the geodesics differs along different directions, indicating the existence of non-constant curvature on the manifold and advocating for estimating the Riemannian metric tensor field instead of applying a single global distance metric.

## 7 HIGH-DIMENSIONAL DATA: AN EXAMPLE WITH MNIST

The curse of dimensionality is a big challenge to apply nonparametric methods to data sources like images and audios. However, it is often found that apparent high-dimensional data actually lie close to some low-dimensional manifold, which is utilized by manifold learning literature to produce reasonable low-dimensional coordinate representations. The proposed method can then be applied to the resulting low-dimensional coordinates as in the following MNIST example.

We embed images in MNIST to a 2-dimensional space via tSNE (Hinton & Roweis, 2002). Similarity between the objects was computed by the sum of the Wasserstein distance between images[4] and the indicator of whether the underlying digits are different (1) or not (0). The goal is to infer the geometry of the embedded data induced by this similarity measures.

The geodesics estimated from our method tend to minimize the number of switches between labels. For example, the geodesic A in Figure 7.1 remains "4" (1st row of panel (b)) throughout, while the straight line on the tSNE chart translates to a path of images switching between "4" and "9"

---

[3]https://www1.nyc.gov/site/tlc/about/tlc-trip-record-data.page. Data format changed after our download.

[4]After rescaling, see Subsection C.4.

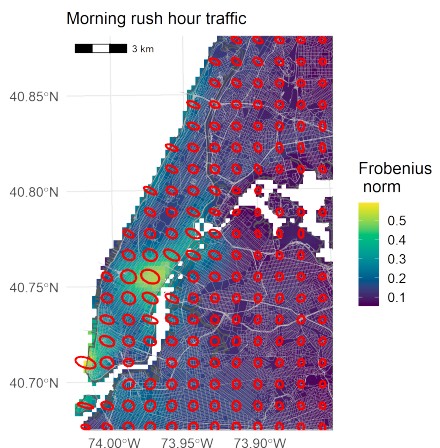

Morning rush hour traffic

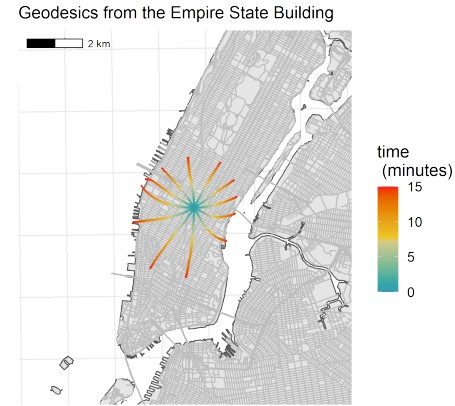

Geodesics from the Empire State Building

(a) Estimated metric tensors for trip duration: cost ellipses and Frobenius norm (background color).

(b) Geodesics correspond to 15-minute taxi rides from the Empire State Building heading to 1–12 o'clock.

Figure 6.1: New York taxi travel time during morning rush hours.

(2nd row of panel (b)); similar phenomenon occurs for geodesics B and C (corresponding to 4th and 7th rows in (b)). Also, our estimated geodesics produce reasonable transition and reside in the space of digits, while unrestricted optimal transport (3rd, 6th, and 9th rows of panel (b)) could produce unrecognizable intermediate images. Our estimated geodesic is faithful to the geometric interpretation that a geodesic is locally the shortest path.

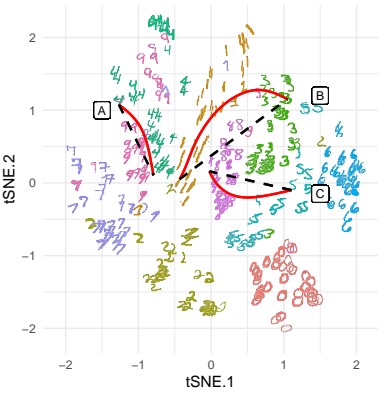

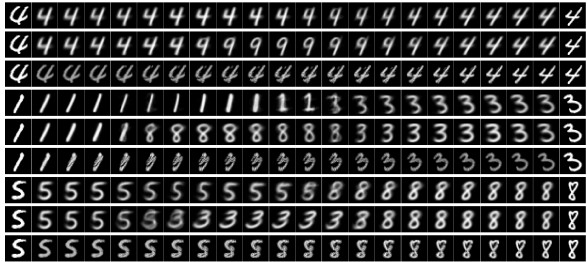

(b) Image transitions corresponding to A, B, and C in (a). Every 3 rows correspond to a set of paths sharing the same pair of starting and ending images, where the first, second, and third rows correspond to the estimated geodesics, the straight lines on the chart, and the optimal transport (path not shown in (a)), respectively.

(a) The estimated geodesic curves (solid red) and straight lines on the chart (dashed black).

Figure 7.1: Geometry induced by a sum of Wasserstein distance and same-digit-or-not indicator.

# 8 DISCUSSION

We present a novel framework for inferring the data geometry based on pairwise similarity measures. Our framework targets data lying on a low-dimensional manifold since observations need to be dense near the locations where we wish to estimate the metric. However, these assumptions are general enough for our method to be applied to manifold data with high ambient dimension in combination with manifold embedding tools. Context-specific interpretation of the geometrical notions, e.g., Riemannian metric and geodesics has been demonstrated in the taxi travel and MNIST digit examples. Our method has the potential to be applied in many other application communities, such as cognition and perception research, where psychometric similarity measures are commonly made.

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

# Supplementary Materials

CONTENTS

## A    ADDITIONAL DEFINITION

A *cost ellipse* visualizes the metric by an ellipse

$$\mathscr{E}_p = \left\{ \left(x^1, \ldots, x^d\right) : \sum_{i,j=1}^{d} \left(x^i - p^i\right) \left(x^j - p^j\right) G^{ij} = r^2 \right\} \qquad (A.1)$$

for some constant $r > 0$, which shows, approximately, the intrinsic distance on the manifold when traveling a unit length on the coordinate chart along each direction. More precisely, it shows the norm of tangent vectors $v^i \partial_i \in T_p \mathcal{M}$ subject to $\sum_{i=1}^d v^i v^i = r^2$ at $p$ pointing to the corresponding direction. For example in a $d = 2$-dimensional manifold at $p = 0$, under $G = \mathrm{diag}\,(\lambda_1, \lambda_2)$ with $\lambda_1 > \lambda_2$ and $r = 1$. The long axis, $(\pm\sqrt{\lambda_1}, 0)$, is the norm of the tangent vector $\pm\partial_1$. Thus, the direction in which the ellipse is larger corresponds to the direction of the larger geodesic distance. One can see (A.1) as the "inverse" of the Tissot's indicatrix, where the latter shows a local equidistance contour to the ellipse's center.

*Frobenius norm* of tensors is denoted as $\|\cdot\|_F$, defined as

$$\|G\|_F = \left( \sum_{i,j=1}^d G_{ij} G_{ij} \right)^{1/2}, \quad \|\Gamma\|_F = \left( \sum_{i,j,k=1}^d \Gamma_{ij}^k \Gamma_{ij}^k \right)^{1/2}, \tag{A.2}$$

for metric tensor $G$ and Christoffel symbol $\Gamma$.

## B  IMPLEMENTATION NOTES

An `R` (R Core Team, 2022) package is developed to implement the proposed methods and all numerical experiments.

We utilized an efficient procedure to obtain estimates $\hat{G}_{ij}$ over the entire manifold $\mathcal{M}$ as follows. We first obtain estimates $\hat{G}_{ij}(p_n)$ over a dense grid of points $p_1, p_2, \ldots, p_{N_{\mathrm{grid}}} \in \mathcal{M}$ by following (3.7)–(3.9). Next, the estimate $\hat{G}_{ij}(x)$ at an arbitrary $x \in \mathcal{M}$ is obtained by the post-smoothing estimate

$$\hat{G}_{ij}(x) = \frac{\sum_{n=1}^{N_{\mathrm{grid}}} K\left(\|x - p_n\| / h_{\mathrm{ps}}\right) \hat{G}_{ij}(p_n)}{\sum_{n=1}^{N_{\mathrm{grid}}} K\left(\|x - p_n\| / h_{\mathrm{ps}}\right)},$$

for some kernel $K$ and bandwidth $h_{\mathrm{ps}} > 0$. We also use local regression (Loader, 1999) for post-smoothing. The grid for the examples (Section 5, Section 6, and Section 7) are $128 \times 128$ for the unit sphere, and $80 \times 80$ for the double spirals, $250 \times 250$ meters cells for the New York taxi example, and $64 \times 64$ for the MNIST example.

The estimated geodesics are computed by numerically solving ordinary differential equations system, either given the start point and initial velocity, or given the start and the end points. It suffices to notice that the geodesic equations (2.2) are equivalently written as, after plugging-in the estimated Christoffel symbol $\hat{\Gamma}$,

$$v^i(t) = \dot{\gamma}^i(t),$$
$$\dot{v}^k(t) = -v^i(t) v^j(t) \hat{\Gamma}_{ij}^k \circ \gamma(t),$$

for $i, j, k = 1, \ldots, d$. Here, $\hat{\Gamma}_{ij}^k \circ \gamma(t)$ is the value of the estimated Christoffel symbol at point $\left(\gamma^1(t), \ldots, \gamma^d(t)\right)$. Further supplying initial condition $\gamma^i(0) = p_0^i$, $v^i(0) = v_0^i$, $i = 1, \ldots, d$ for point $p_0 \in \mathcal{M}$ and tangent vector $v_0 \in T_{p_0}\mathcal{M}$ constitute an initial value problem, whose solution reflects the geodesic curve starting from $p_0$ with initial velocity $v_0$. On the other hand, supplying boundary condition $\gamma^i(0) = p_0^i$, $\gamma^i(1) = p_1^i$, $i = 1, \ldots, d$ for $p_0, p_1 \in \mathcal{M}$ constitute a boundary value problem, whose solution reflects the geodesic curve from $p_0$ to $p_1$. we use `deSolve` (Soetaert et al., 2010) and `bvpSolve` (Mazzia et al., 2014) for initial value problems and boundary value problems respectively. See reference therein for further details of numeric solution to ODE.

## C  ADDITIONAL EXPERIMENT DETAILS

This section provides further detail to complete Section 5, Section 6, and Section 7 of the main text including how we generated the simulated data, and more figures.

## C.1    UNIT SPHERE AND ROUND METRIC

The stereographic coordinate of the $d$-dimensional sphere $\mathbb{S}^d$ identifies points on the sphere by mapping it to its stereographic projection in $\mathbb{R}^d$ from the north pole. The round metric on the sphere $\mathbb{S}^d$ is the metric induced by embedding of $\mathbb{S}^d \hookrightarrow \mathbb{R}^{d+1}$. For detail, see for example page 30 of Lee (2013) and chapter 3 of Lee (2018). We generated endpoints $X_{u0}, X_{u1}$ uniformly in the coordinate chart $(-3, 3) \times (-3, 3)$, then pair the endpoints so that the difference in coordinates of the endpoints $|\delta_{u0}^i - \delta_{u1}^i| = |X_{u0}^i - X_{u1}^i|$ would not exceed 0.2 for $i = 1, 2$, $u = 1, \ldots, N$.

More precisely, data were generated on $d = 2$-dimensional sphere under stereographic projection coordinate. A total of $N = 5 \times 10^5$ pairs of endpoints with $X_{u0}^i, X_{u1}^i \in (-3, 3)$ were generated subject to $|X_{u0}^i - X_{u1}^i| \le 0.2$ for all $u = 1, \ldots, N; i = 1, \ldots, d$. For a reasonable signal-to-noise ratio, we set $\sigma(p) = \sigma \approx 9 \times 10^{-4}$ for all $p$, which is approximately one-tenth of the marginal expectation of squared distance, i.e., $\sigma \approx \mathbb{E}\, dist\, (X_{u0}, X_{u1})^2 / 10$.

For simplicity of presentation, we scaled the distance for the binary similarity response model (3.3). More precisely, we use $dist_c (\cdot, \cdot) = \sqrt{c}\, dist\, (\cdot, \cdot)$ induced by the scaled metric $G_{ij,c} = c\mathring{G}_{ij}$ for some constant $c$ and $i, j = 1, \ldots, d$. The experiment here used $c = 300$. Intuitively, the constant $c$ regulates the signal-to-noise ratio without changing the form of geodesics. Given the endpoints, a smaller $c$ leads to a smaller value of geodesic distance and hence smaller variation in the linear predictors $\eta_u$, so the response $Y_u$ will take less influence from the distance, representing a higher amount of noise. Then $\hbar(p)$ was set to be the average local squared distances within a local neighborhood of $p$ under the scaled distance.

In the end, the responses were generated following (3.2) and (3.3) respectively.

In addition, Figure C.1 illustrates the relative Frobenius error for estimated tensors using noiseless or binary responses.

### C.1.1    BANDWIDTH SELECTION

Like local regression, the proposed method relies on a neighborhood specification for optimal bias-variance trade-off. The simulation in Subsection 5.1 uses the rectangular kernel $K(x) = \mathbf{1}_{[-1,1]}(x)$ for (3.10), where $\mathbf{1}$ is the indicator function, so the estimation only utilizes observations with endpoints $X_{u0}, X_{u1}$ are both lying in the neighborhood $\mathcal{U}_p = \left\{ (x^1, \ldots, x^d) : |x^i - p^i| \le h, i = 1, \ldots, d \right\}$ of the target point $p$.

We propose a train–test set scheme for data-driven bandwidth selection. To simplify computation, we only considered additive error under (3.2). A $16 \times 16$ grid $p_1, \ldots, p_{256} \in (-3, 3) \times (-3, 3)$ were used as target points where metric tensors were estimated, with a test set of $N_{\text{test}} = 31246$ observations that were within close proximity to the grid. Estimation of the tensors were computed w.r.t. bandwidth $h$ utilizing a train set containing $N_{\text{train}} = 400158$ (approximately 80% of the data) randomly selected observations outside of the test set. For the test set, $\hat{Y}_{u,\text{test}} = \hat{\eta}_{u,\text{test}}$ were then computed under identity link by plugging the estimated tensors into (3.6).

The bandwidth minimizing the squared loss $\sum_{u=1}^{N_{\text{test}}} \left( Y_{u,\text{test}} - \hat{Y}_{u,\text{test}} \right)^2 / N_{\text{test}}$ is then chosen. The proposed bandwidth selection resulted in $h = 0.18$ according to the loss shown in the left panel of Figure C.2, which corresponds to an effectively local sample size of approximately 1000 in the training step (right panel of Figure C.2). These tuning parameter choices were applied in the results shown in Subsection 5.1. Other bandwidth selection methods developed for local regression (see e.g. Fan & Gijbels, 1996, section 4.10) can also be adopted here.

### C.2    THE DOUBLE SPIRALS

Define a class of spiral functions as $\mathcal{S} : \mathbb{R} \to \mathbb{R}^2$ with

$$t \mapsto (\cos(5t + \phi),\ t\sin(5t + \phi)).$$

The underlying spirals for class $\mathcal{A}$ and $\mathcal{B}$ are $\mathcal{S}_A = \mathcal{S}(\cdot, \phi = 0)$ and $\mathcal{S}_B = \mathcal{S}(\cdot, \phi = \pi)$ respectively. For endpoints, we first generate points $\mathring{X}_m$ independently and uniformly on $\mathcal{S}_A$ or $\mathcal{S}_B$, then the endpoints are generated following $\mathring{X}_m + 0.15 Z_m$ for i.i.d. standard Gaussian random variables

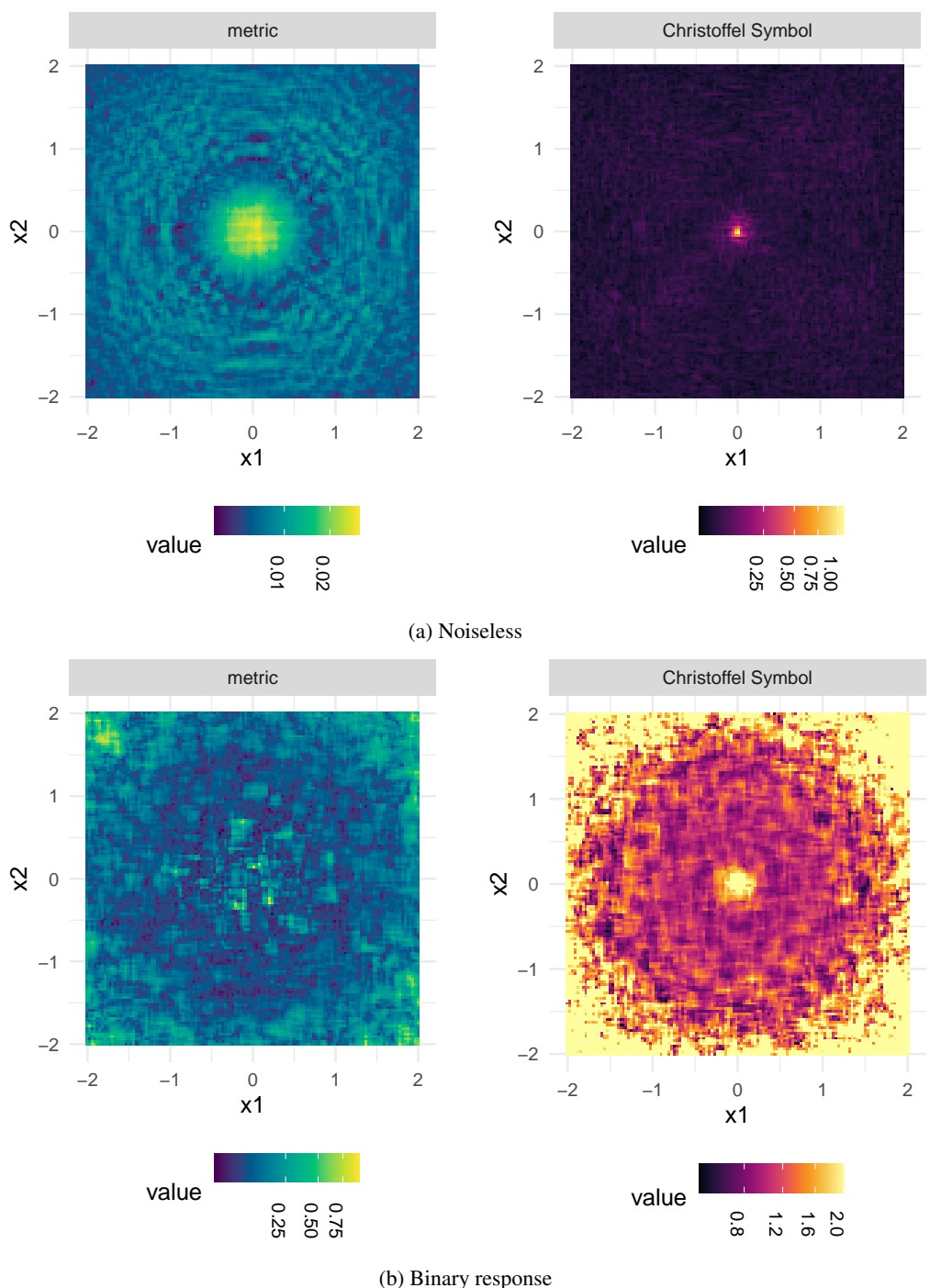

Figure C.1: Relative errors w.r.t. Frobenius norm (A.2) of the estimated tensors with noiseless or binary response for 2-dimensional sphere under stereographic projection coordinate chart.

$Z_m$, $m = 1, \ldots, 70000$. Provided with those candidate endpoints, we pair them to form relative comparison subject to the restriction that $|X_{u0}^i - X_{uj}^i| \le 0.35$ for $i, j = 1, 2$, $n = 1, \ldots, N$. The responses $Y_u$ are then generated based on the class of involving endpoints by their corresponding $\mathring{X}$ on the spirals.

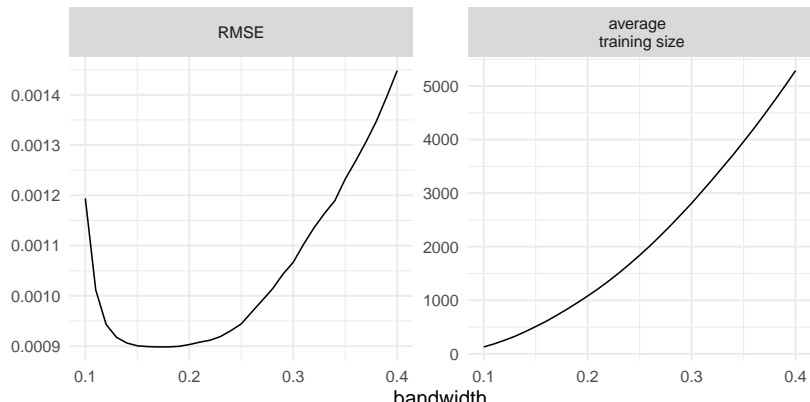

Figure C.2: Root mean squared error (RMSE) for the test set (Left) and the average number of local training observations (Right) as the bandwidth $h$ varies.

For estimation, we used a larger local neighborhood $\mathcal{U}_p = \left\{ (x^1, x^2) : |x^i - p^i| \leq h \text{ for } i = 1, 2 \right\}$ with $h = \pi/2$ and weights $w_u = 1_{\{X_{u0}, X_{u1}, X_{u2} \in \mathcal{U}_p\}}$ for $u = 1, \dots, N$ to avoid degenerate estimates.

Note that different starting points and initial velocities will generate different geodesics, not all resembling a spiral, as shown in Figure C.3.

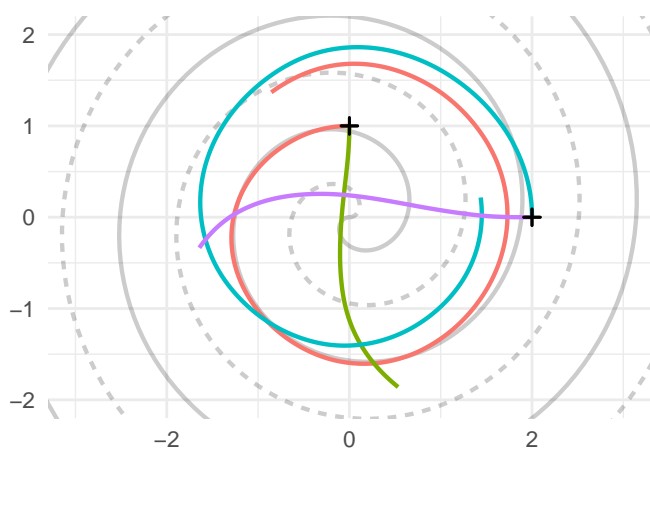

Figure C.3: Geodesics with different starting points and initial velocities under estimated metric, crosses indicate starting points.

## C.3 NYC TAXI TRIPS

We focus on the 8,809,982 sensible records between 7 a.m. to 10 a.m. on business days from May to September (summer months to hopefully avoid snow) of 2015 in New York City areas other than the Staten Island. Sensible in terms of GPS coordinates not falling in to the rivers, travel time not being several seconds, and that inferred traveling speed is not 120 mph, and e.t.c. We measure the cost to travel $Y_u$ by the squared trip duration (instead of the trip distance). For each target location $p$, estimation was computed using trips among the $M \leq 5 \times 10^4$ closest pickup/dropoff endpoints

in the neighborhood $\mathcal{U}_p = \left\{ (x^1, x^2) : |x^i - p^i| \leq 5 \text{ kilometers for } i = 1, 2 \right\}$, and weights given by $h = 2.5$ kilometers with the kernel $K$ being the density function of the standard normal distribution.

## C.4 THE MNIST EXAMPLE

The dimension reduction is computed using R package `dimRed` (Kraemer et al., 2018). The Wasserstein distance and optimal transport are computed using package `transport` (Schuhmacher et al., 2022). To show image transitions, weighted average is adopted to approximate the inverse of the tSNE embedding so as to map the trajectories back to image space, similar to, e.g., equation (3.9) of Chen & Müller (2012), but with Gaussian kernel and a sufficiently small bandwidth.

To simplify computation, we only embed the first $3 \times 10^4$ images (half of the entire data), and the resulting embedding coordinates were scaled (i.e., centered by the mean and divided by standard deviation). We generated $N = 10^5$ comparison by selecting nearby points in the embedded space subject to $\|X_{u0} - X_{u1}\|_\infty \leq 0.75$, whose response $Y_u$ were computed based on the same-digit-or-not indicator and 2–Wasserstein distance between the corresponding images:

$$dist\left( X_{u0}, X_{u0} \right) = C dist_{wass} \left( \text{pic}_{u0}, \text{pic}_{u1} \right) + \mathbb{1}_{\{\text{lbl}_{u0} \neq \text{lbl}_{u1}\}},$$

where for $u = 1, \ldots, N$,

- $X_{u0}, X_{u1} \in \mathbb{R}^2$ are coordinates in the embedded space;
- $\text{pic}_{u0}$ and $\text{pic}_{u0}$ are the $28 \times 28$ grey scale images;
- $\text{lbl}_{u0}$ and $\text{lbl}_{u1}$ are the image labels (0–9);
- $dist_{wass} \left( \cdot, \cdot \right)$ is the 2–Wasserstein distance treating images as 2-dimensional probability distributions;
- $\mathbb{1}_{\{\text{event}\}}$ is the indicator for whether the event is true (1) or false (0).

We multiply the Wasserstein distance by $C = 4$ to balance the magnitude of the two summands, otherwise the later could be overly dominating.

Estimation was drawn under model (3.2) with squared loss $Q(y, \mu) = (y - \mu)^2$ and the identity link. We included the intercept term ($\beta^{(0)}$ in (3.6)) to capture intrinsic variation. Since the dimensional reduction embedding map is not necessarily an injection, so that different images with non-zero similarity measures could share identical coordinates in the embedded space. Figure C.4a shows the estimated intercepts, which is larger among class boundaries, coherent to a greater variation in the similarity measure. Those few blank pixels indicate failure to obtain positive definite metric, which are alleviated by averaging neighboring estimated values.

We also dropped the terms for Christoffel symbols from (3.6) for better numeric stability. Consequently, the estimated Christoffel symbols were computed by numeric differential following the definition. Results are similar if we include the Christoffel symbol terms in the linear predictor, but less stable.

Figure C.4b shows the cost ellipses for addition visualization.

Notably, the proposal also work supplied with binary similarity measures using only the same-digit-or-not indicator (i.e., setting $C = 0$ to remove the Wasserstein distance), and retains the "fewer label switching" tendency as illustrated in Figure C.5. We see this as an real data example in analogy to the double spirals (Subsection 5.2). We would also like to remark that not all geodesics are different from straight lines on the chart, and it is not guaranteed that geodesic must travel within the same class whenever possible, since its travel is jointly determined by the metric and where the endpoints are located.

## D SPREAD OF GEODESICS

Here we provide proof to the Proposition 3.1 in the main text, which characterizes the distance between geodesics departing from a same starting point. Proposition 3.1 is a result of combining Proposition D.1 and Proposition D.2.

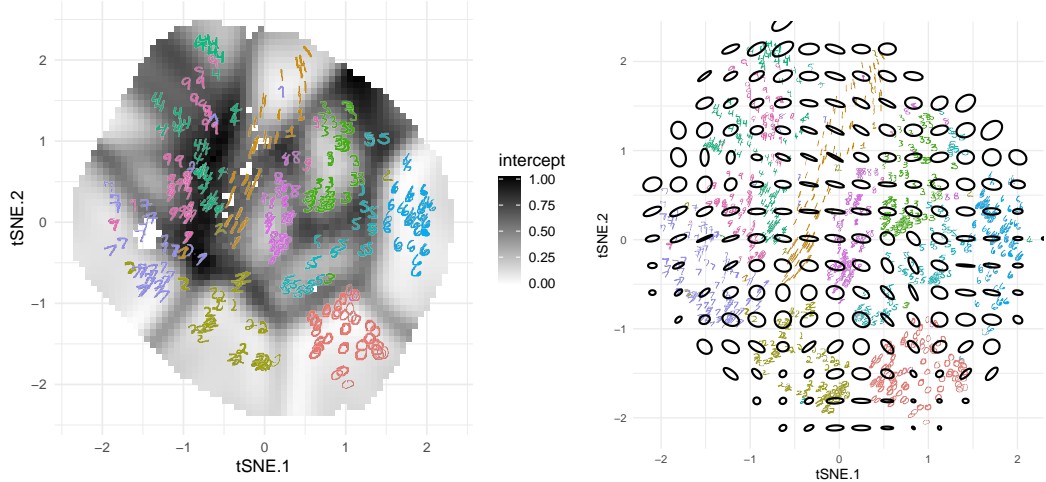

(b) The cost ellipses of estimated metric.

(a) Estimated intercept reflecting intrinsic local variation of the similarity measure.

Figure C.4: More figures for the induced geometry by adding Wasserstein distance and same-digit-or-not indicator.

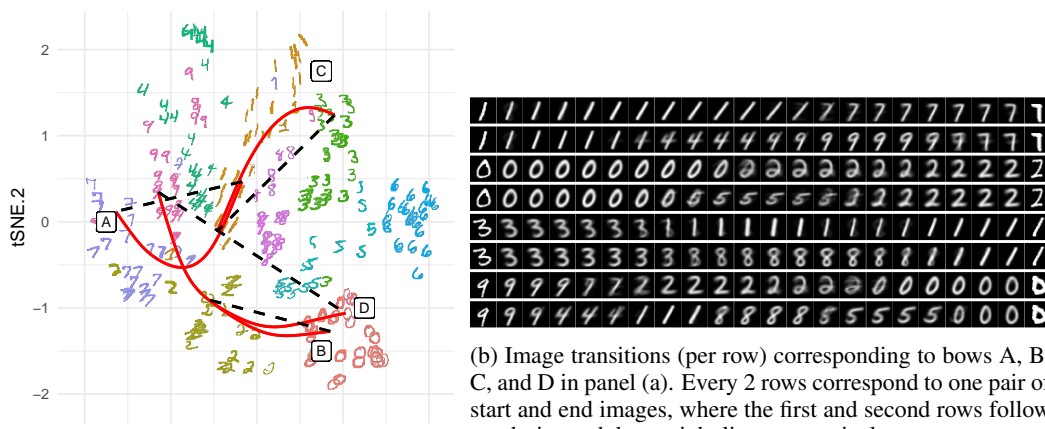

(b) Image transitions (per row) corresponding to bows A, B, C, and D in panel (a). Every 2 rows correspond to one pair of start and end images, where the first and second rows follow geodesics and the straight lines respectively.

(a) The geodesic curves (solid red) and straight lines (dashed black).

Figure C.5: Induced geodesics of only the same-digit-or-not indicator.

*Proposition* D.1 (spread of geodesics). Let $p \in \mathcal{M}$ and $v, w \in T_p\mathcal{M}$ be two tangent vector at $p$. Then the squared distance of separation satisfies Taylor expansion of

$$dist\left(\exp_p(tv), \exp_p(tw)\right)^2 = t^2 \|v - w\|^2 - \frac{1}{3}t^4 \langle R(v, w)w, v \rangle + O(t^5)$$

as $t \to 0$.

Here, $R$ is the $(1, 3)$-*curvature tensor* defined as

$$R(X, Y)Z = \nabla_X \nabla_Y Z - \nabla_Y \nabla_X Z - \nabla_{[X,Y]}Z,$$

where $X, Y, Z$ are vector fields and $[X, Y] = XY - YX$ is the *Lie bracket* (c.f., e.g., Lee, 2018, page 385). Further, the *Riemann curvature tensor* is defined as

$$Rm(X, Y, Z, W) = \langle R(X, Y)Z, W \rangle,$$

where $W$ is also a vector field. Note that $R$ and $Rm$ are both tensor fields, so $\langle R(v, w)w, v \rangle$ (equivalently $Rm(v, w, w, v)$) are those evaluated at $p$, since $v, w \in T_p\mathcal{M}$. See Lee (2018), pp. 196–199 for detail.

However, additional terms are introduced when computing via coordinate charts, as a result of approximating the initial velocities $v$ and $w$.

*Proposition* D.2 (approximation of velocity). For any $p \in \mathcal{M}$, let $v \in T_p\mathcal{M}$ be a tangent vector at $p$ and $\gamma(t) = \exp_p(tv)$ be the geodesic from $p$ with initial velocity $v$. Given any local coordinate chart, write $v = v^i \partial_i$. For $i = 1, \ldots, d$, denote $\delta^i = \delta^i(t) = \gamma^i(t) - \gamma^i(0)$ as the difference in coordinate after traveling $t$ along $\gamma$, we have

$$v^i = t^{-1}\delta^i(t) + \mathscr{R}^i(t),$$

where the remainder is

$$\mathscr{R}^i(t) = \frac{1}{2t}\delta^m\delta^n\Gamma^i_{mn} + \frac{1}{6t}\delta^m\delta^n\delta^l\left(\Gamma^k_{mn}\Gamma^i_{kl} + \partial_l\Gamma^i_{mn}\right) + O(t^3) \tag{D.1}$$

as $t \to 0$, where $\Gamma$ and $\partial\Gamma$ denote the the Christoffel symbols and their derivatives at $p$.

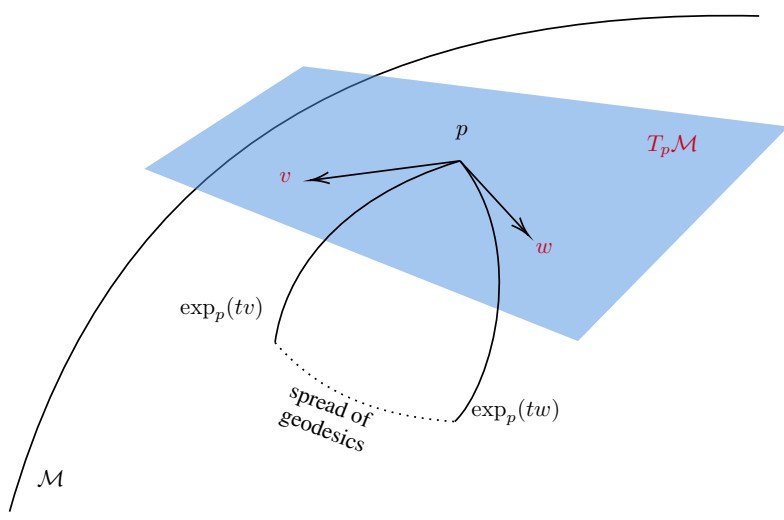

Figure D.1: A visualization for the spread of geodesics as in Proposition 3.1. A tangent space (blue plane) and tangent vectors are annotated in red.

## D.1 PROOFS

*Proof of Proposition D.1.* Similar results can be found at Proposition 2.7 of do Carmo (1992), Proposition 5.4, of (Lang, 1999, IX, §5). We use the form presented by Meyer (1989). In the following, we reproduce the proof to the equation (9) of Meyer (1989) with some additional clarification.

Let $\gamma_0(s) = \exp_p(sv)$ and $\gamma_1(s) = \exp_p(sw)$, define a family of curves

$$V(s, t) = \exp_{\gamma_0(s)}\left(t \exp^{-1}_{\gamma_0(s)} \gamma_1(s)\right),$$

so that the curves $V_s : t \mapsto V(s, t)$ are geodesics connecting $\gamma_0(s)$ and $\gamma_1(s)$ (c.f., e.g., proposition 5.19 and equation (10.2) of Lee (2018)), and that $V$ is a variation through geodesics $V_s$. Further, let $T = \partial_t V$, which is a tangent field of velocities. Let $E = \partial_s V$, which is a Jacobi field through geodesics $V_s$ that vanishes at $p$. Denote $H(s) = dist\left(\gamma_0(s), \gamma_1(s)\right)^2 = \|T\|^2|_{s,t}$ for any $t \in [0, 1]$,

where the "$|_{s,t}$" means to take value at point $V(s,t)$. Then by the chain rules for covariant derivatives (see, e.g. Lee, 2018, chapter 4), we have

$$\frac{d}{ds}H(s) = \frac{d}{ds}\langle T,T\rangle|_{s,t} = 2\langle D_sT,T\rangle|_{s,t},$$

$$\left(\frac{d}{ds}\right)^2 H(s) = 2\left(\langle D_s^2T,T\rangle + \|D_sT\|^2\right)|_{s,t},$$

$$\left(\frac{d}{ds}\right)^3 H(s) = 2\left(\langle D_s^3T,T\rangle + 3\langle D_s^2T,D_sT\rangle\right)|_{s,t},$$

$$\left(\frac{d}{ds}\right)^4 H(s) = 2\left(\langle D_s^4T,T\rangle + 3\left\|D_s^2T\right\| + 4\langle D_s^3T,D_sT\rangle\right)|_{s,t}.$$

Note that $V_0 = p$ for all $t$, so that $T|_{s=0,t} = 0$ for all $t$, hence $H'(0) = 0$.

Note that $V_s : t \mapsto V(s,t)$, $s \mapsto V(s,0)$ and $s \mapsto V(s,1)$ are geodesics; thus $D_tT = 0$ for all $t$, $D_sE|_{s,t=0} = 0$ and $D_sE|_{s,t=1} = 0$ for all $s$. In addition, by lemma 6.2 of Lee (2018), $D_sT = D_tE$.

By Jacobi equation, $D_t^2E + R(E,T)T = 0$ for all $s$, which implies $D_t^2E|_{s=0} = 0$ since $T|_{s=0} = 0$. This means the vector field $t \mapsto E|_{s=0,t}$ at $p$ is linear in $t$, together with $E|_{s=0,t=0} = v$ and $E|_{s=0,t=1} = w$, we can write

$$E|_{s=0,t} = v + t(w - v)$$

for $t \in [0,1]$. Therefore $D_sT|_{s=0,t} = D_tE|_{s=0,t} = w - v$, which implies $H''(0) = 2\|v - w\|^2$.

Proceeding to the third order derivatives, observe that

$$H'''(0) = 6\langle D_s^2T,D_sT\rangle|_{s=0,t},$$

and by proposition 7.5 of Lee (2018), $D_s^2T = D_sD_tE = D_tD_sE + R(E,T)E$, thus it suffices to show

$$D_sE|_{s=0,t} = 0, \text{ for all } t, \tag{D.2}$$

in order to argue $H'''(0) = 0$. Since it is known that $D_sE|_{s=0,t=0} = 0 = D_sE|_{s=0,t=1}$, it suffices to consider its derivative for (D.2). Use proposition 7.5 of Lee (2018) repeatedly, we have

$$D_t^2D_sE|_{s=0,t}$$
$$= D_tD_sD_tE|_{s=0,t} + D_t\left(R(T,E)E\right)|_{s=0,t}$$
$$= D_tD_s^2T|_{s=0,t}$$
$$= \left(D_sD_tD_sT - R(E,T)(D_sT)\right)|_{s=0,t}$$
$$= D_sD_tD_sT|_{s=0,t}$$
$$= D_s\left(D_sD_tT - R(E,T)T\right)|_{s=0,t}$$
$$= D_s\left(R(T,E)T\right)|_{s=0,t},$$

where the last equation is due to $D_tT = 0$. Further, by chain rule of covariant derivative (c.f. e.g. proposition 4.15 of Lee (2018)),

$$D_s\left(R(T,E)T\right) = \left(\nabla_E R\right)(T,E)T + R(D_sT,E)T + R(T,D_sE)T + R(T,E)D_sT,$$

which equals to zero at $s = 0, t$ since $T|_{s=0,t} = 0$ for all $t$. Hence $t \mapsto D_sE|_{s=0,t}$ is also a linear vector field, implying (D.2) and subsequently $H'''(0) = 0$.

For the fourth order derivative, note that (D.2) also implies that $D_tD_sE|_{s=0,t} = 0$ and that $D_s^2T|_{s=0,t} = 0$ for all $t$. Therefore,

$$H^{(4)}(0) = 8\langle D_s^3T,D_sT\rangle|_{s=0,t}.$$

Further,

$$D_s^3T = D_s^2D_tE$$
$$= D_s\left(D_tD_sE + R(E,T)E\right)$$
$$= D_sD_tD_sE + \left(\nabla_E R\right)(E,T)E + R(D_sE,T)E + R(E,D_sT)E + R(E,T)(D_sE),$$

so $D_s^3 T|_{s=0,t} = (D_s D_t D_s E + R(E, D_s T)E)|_{s=0,t}$. Thus,

$$\left\langle D_s^3 T, D_s T \right\rangle |_{s=0,t} = \left( \langle D_s D_t D_s E, D_s T \rangle + \langle R(E, D_s T)E, D_s T \rangle \right)|_{s=0,t}.$$

Recall at $s = 0$, $D_s T|_{s=0,t} = D_t E|_{s=0,t} = w - v$, therefore

$$\begin{aligned}
& \langle R(E, D_s T)E, D_s T \rangle |_{s=0,t} \\
&= Rm(E, D_s T, E, D_s T)|_{s=0,t} \\
&= Rm(v, w - v, v, w - v) + tRm(v, w - v, w - v, w - v) \\
&= Rm(v, w, v, w).
\end{aligned}$$

Further,

$$\begin{aligned}
& \langle D_s D_t D_s E, D_s T \rangle |_{s=0,t} \\
&= \left( D_s \langle D_t D_s E, D_s T \rangle - \langle D_t D_s E, D_s^2 T \rangle \right)|_{s=0,t} \\
&= D_s \langle D_t D_s E, D_s T \rangle |_{s=0,t} \\
&= D_s D_t \langle D_s E, D_s T \rangle |_{s=0,t} - D_s \langle D_s E, D_t^2 E \rangle |_{s=0,t},
\end{aligned}$$

where the second term in the last line vanishes since $D_t^2 E|_{s=0,t} = 0$ and $D_s E|_{s=0,t} = 0$. Moreover, since the Levi–Civita connection is torsion free, we have

$$\langle D_s D_t D_s E, D_s T \rangle |_{s=0,t} = D_s D_t \langle D_s E, D_s T \rangle |_{s=0,t} = D_t D_s \langle D_s E, D_s T \rangle |_{s=0,t},$$

which should be irrelevant to $t$, so that $D_s \langle D_s E, D_s T \rangle |_{s=0,t}$ is linear in $t$. Yet

$$D_s \langle D_s E, D_s T \rangle = \left\langle D_s^2 E, D_s T \right\rangle + \left\langle D_s E, D_s^2 T \right\rangle,$$

which vanishes at $s = 0$ and $t = 0, 1$. Hence $D_s \langle D_s E, D_s T \rangle |_{s=0,t} = 0$ for all $t \in [0,1]$. Combining those with the symmetries of Riemann curvature tensor leads to the desired expansion. □

*Proof of Proposition D.2.* Under the coordinate chart, we can write the geodesic curve as $\gamma : t \mapsto \left( \gamma^1(t), \ldots, \gamma^d(t) \right)$ for some smooth function $\gamma^1, \ldots, \gamma^d$. Then for any $i = 1, \ldots, d$, univariate Taylor expansion provides

$$\gamma^i(t) = \gamma^i(0) + \dot{\gamma}^i(0)t + \frac{1}{2}t^2 \ddot{\gamma}^i(0) + \frac{1}{6}t^3 \dddot{\gamma}^i(0) + O(t^4)$$

as $t \to 0$, where $\dot{\gamma}^i$, $\ddot{\gamma}^i$, and $\dddot{\gamma}^i$ are the first, second, and third order derivative of $\gamma^i$ w.r.t. $t$. Note that the first derivative $\dot{\gamma}^i(0) = v^i$, and the geodesic equation and its derivative give

$$\begin{aligned}
\ddot{\gamma}^i(0) &= -v^m v^n \Gamma^i_{mn}, \\
\dddot{\gamma}^i(0) &= v^m v^n v^l \left( 2\Gamma^k_{mn}\Gamma^i_{kl} - \partial_l \Gamma^i_{mn} \right).
\end{aligned}$$

Plugging into the initial Taylor expansion gives the desired result. □

*Proof of Proposition 3.1 in the maintext.* By Proposition D.1 and Proposition D.2, as $t \to 0$, we have

$$\begin{aligned}
& t^2 \|v - w\|^2 \\
&= \left( \delta^i_{0-1} + t\mathscr{R}^i_0(t) - t\mathscr{R}^i_1(t) \right) G_{ij} \left( \delta^j_{0-1} + t\mathscr{R}^j_0(t) - t\mathscr{R}^j_1(t) \right) \\
&= \delta^i_{0-1}\delta^j_{0-1} G_{ij} + 2t\delta^i_{0-1} \left( \mathscr{R}^j_0(t) - \mathscr{R}^j_1(t) \right) G_{ij} + O(t^4) \\
&= \delta^i_{0-1}\delta^j_{0-1} G_{ij} + \delta^i_{0-1} \left( \delta^k_0 \delta^l_0 - \delta^k_1 \delta^l_1 \right) \left( \Gamma^j_{kl} G_{ij} \right) + O(t^4),
\end{aligned}$$

where

$$\mathscr{R}^i_a(t) = \frac{1}{2t}\delta^m_a \delta^n_a \Gamma^i_{mn} + \frac{1}{6t}\delta^m_a \delta^n_a \delta^l_a \left( \Gamma^k_{mn}\Gamma^i_{kl} + \partial_l \Gamma^i_{mn} \right) + O(t^3)$$

for $a = 0, 1$, similar to (D.1). Note that $\delta^i_0 = \delta^i_0(t) = O(t)$, it suffices to keep only the first term in the $\mathscr{R}^j_a(t)$, which is $O(t)$. □

# E  ASYMPTOTIC OF THE ESTIMATED METRIC TENSOR

Now we discuss the variance and bias of the estimated metric tensors. For simplicity, use the squared loss $Q(\mu, y) = (\mu - y)^2$, the identity link $g(\mu) = \mu$, and exclude the intercept $\beta^{(0)}$ and the terms $\beta^{(2)}_{ijk}$ for derivative. Given a suitable order of the indices $i, j$, we rewrite (3.6) into matrix form. Denote

$$\mathbf{D}_u = \left(\delta^1_{u,0-1}\delta^1_{u,0-1}, \ldots, \delta^i_{u,0-1}\delta^j_{u,0-1}, \ldots, \delta^d_{u,0-1}\delta^d_{u,0-1}\right)^T,$$

$$\boldsymbol{\beta} = \left(\beta^{(1)}_{11}, \ldots, \beta^{(1)}_{ij}, \ldots, \beta^{(1)}_{dd}\right)^T,$$

then the linear predictor $\eta_n = \mathbf{D}_u^T \boldsymbol{\beta}$. Further, write

$$\mathbf{D} = (\mathbf{D}_1, \ldots, \mathbf{D}_N)^T, \quad \boldsymbol{Y} = (Y_1, \ldots, Y_N)^T, \quad \mathbf{W} = \mathrm{diag}\,(w_1, \ldots, w_N),$$

so the loss (3.7) becomes

$$(\boldsymbol{Y} - \mathbf{D}\boldsymbol{\beta})^T \mathbf{W} (\boldsymbol{Y} - \mathbf{D}\boldsymbol{\beta}), \tag{E.1}$$

whose minimizer is $\hat{\boldsymbol{\beta}} = \left(\mathbf{D}^T\mathbf{W}\mathbf{D}\right)^{-1} \mathbf{D}^T\mathbf{W}\boldsymbol{Y}$. Therefore

$$\mathrm{bias}\left(\hat{\boldsymbol{\beta}}|\mathbf{D}\right) = \left(\mathbf{D}^T\mathbf{W}\mathbf{D}\right)^{-1} \mathbf{D}^T\mathbf{W}\boldsymbol{r},$$

$$\mathrm{var}\left(\hat{\boldsymbol{\beta}}|\mathbf{D}\right) = \left(\mathbf{D}^T\mathbf{W}\mathbf{D}\right)^{-1} \mathbf{D}^T\boldsymbol{\Sigma}\mathbf{D} \left(\mathbf{D}^T\mathbf{W}\mathbf{D}\right)^{-1},$$

where

$$\boldsymbol{r} = \left(\mathbb{E}\,(Y_u|\mathbf{D}) - \delta^i_{u,0-1}\delta^j_{u,0-1}G_{ij}\right)_{1 \leq n \leq N},$$

$$\boldsymbol{\Sigma} = \mathrm{diag}\left(w_u^2\,\mathrm{var}\,(Y_u|X_{u0}, X_{u1})\right)_{1 \leq n \leq N}.$$

The assumptions (A1)–(A4) in the main text are reiterated here.

(A1)  The joint density of endpoints $X_{u0}, X_{u1}$ is positive and continuously differentiable.

(A2)  The functions $G_{ij}, \Gamma^k_{ij}$ are $C^2$-smooth for $i, j, k = 1, \ldots, d$.

(A3)  The kernel $K$ in weights (3.10) is symmetric, continuous, and has bounded support.

(A4)  $\sup_u \mathrm{var}\,(Y_u|X_{u0}, X_{u1}) < \infty$.

*Proposition* E.1.  Denote

$$S_{1N,i_1i_2i_3i_4} = \sum_{u=1}^{N} w_u \delta^{i_1}_{u,0-1}\delta^{i_2}_{u,0-1}\delta^{i_3}_{u,0-1}\delta^{i_4}_{u,0-1},$$

$$S_{2N,i_1i_2i_3i_4} = \sum_{u=1}^{N} w_u^2\,\mathrm{var}\,(Y_u|X_{u0}, X_{u1})\,\delta^{i_1}_{u,0-1}\delta^{i_2}_{u,0-1}\delta^{i_3}_{u,0-1}\delta^{i_4}_{u,0-1},$$

$$S_{3N,i_1i_2} = \sum_{u=1}^{N} w_u \delta^{i_1}_{u,0-1}\delta^{i_2}_{u,0-1}R_u,$$

where

$$R_u = \sum_{1 \leq k,l,m,r \leq d} \delta^m_{u,0-1}\left(\delta^k_{n0}\delta^l_{n0} - \delta^k_{n1}\delta^l_{n1}\right)\Gamma^r_{kl}G_{mr}.$$

Under (A1), (A2), (A3), and (A4), and suppose that $h \to 0$ and $Nh^{2d} \to \infty$, then

$$\mathbb{E}S_{1N,i_1i_2i_3i_4} = O\left(Nh^4\right), \quad \mathrm{var}\,S_{1N,i_1i_2i_3i_4} = O\left(Nh^{8-2d}\right),$$

$$\mathbb{E}S_{2N,i_1i_2i_3i_4} = O\left(Nh^{4-2d}\right), \quad \mathrm{var}\,S_{2N,i_1i_2i_3i_4} = O\left(Nh^{8-6d}\right),$$

$$\mathbb{E}S_{3N,i_1i_2i_3i_4} = O\left(Nh^6\right), \quad \mathrm{var}\,S_{3N,i_1i_2i_3i_4} = O\left(Nh^{10-2d}\right),$$

as $h \to 0$ and $N \to \infty$. So

$$S_{1N,i_1i_2i_3i_4} = O_p\left(Nh^4\right), \quad S_{2N,i_1i_2i_3i_4} = O_p\left(Nh^{4-2d}\right).$$

If further $Nh^{2+2d} \to \infty$, then

$$S_{3N,i_1 i_2 i_3 i_4} = O_p\left(Nh^6\right),$$

as $h \to 0$ and $N \to \infty$.

*Proof.* Write

$$U_{u;i_1 i_2 i_3 i_4} = w_u \delta_{u,0-1}^{i_1} \delta_{u,0-1}^{i_2} \delta_{u,0-1}^{i_3} \delta_{u,0-1}^{i_4},$$

so

$$
\begin{aligned}
&\mathbb{E} U_{u;i_1 i_2 i_3 i_4} \\
&= \int h^{-2d} \prod_{i=1}^{d} \left(K\left(\delta_{n0}^i/h\right) K\left(\delta_{n1}^i/h\right)\right) \delta_{n,0-1}^{i_1} \delta_{n,0-1}^{i_2} \delta_{n,0-1}^{i_3} \delta_{n,0-1}^{i_4} \cdot \\
&\quad f(p^1 + \delta_{n0}^1, \ldots, p^d + \delta_{n1}^d) d\delta_{n0}^1 \ldots d\delta_{n1}^d \\
&= h^4 \int K\left(s_{u0}^1\right) \cdot \ldots \cdot K\left(s_{u1}^d\right) \left(s_{u0}^{i_1} - s_{u1}^{i_1}\right)\left(s_{u0}^{i_2} - s_{u1}^{i_2}\right)\left(s_{u0}^{i_3} - s_{u1}^{i_3}\right)\left(s_{u0}^{i_4} - s_{u1}^{i_4}\right) \cdot \\
&\quad \left(f(p^1, \ldots p^d) + o(1)\right) ds_{u0}^1 \ldots ds_{u1}^d \\
&= O(h^4),
\end{aligned}
$$

where $f$ is the joint density of endpoints $X_{u0}, X_{u1}$, the second last equality is due to change of variables, and the last due to (A1). Similar argument implies

$$
\begin{aligned}
\operatorname{var} U_{u;i_1 i_2 i_3 i_4} &= O\left(h^{8-2d}\right), \\
\mathbb{E} w_u U_{u;i_1 i_2 i_3 i_4} &= O\left(h^{4-2d}\right), \\
\operatorname{var} w_u U_{u;i_1 i_2 i_3 i_4} &= O\left(h^{8-6d}\right).
\end{aligned}
$$

These rates apply uniformly over $n$, therefore by i.i.d. and that $\operatorname{var} Y_u | X_{u0}, X_{u1}$ is uniformly bounded,

$$
\begin{aligned}
\mathbb{E} S_{1N,i_1 i_2 i_3 i_4} &= O\left(Nh^4\right), & \operatorname{var} S_{1N,i_1 i_2 i_3 i_4} &= O\left(Nh^{8-2d}\right), \\
\mathbb{E} S_{2N,i_1 i_2 i_3 i_4} &= O\left(Nh^{4-2d}\right), & \operatorname{var} S_{2N,i_1 i_2 i_3 i_4} &= O\left(Nh^{8-6d}\right).
\end{aligned}
$$

Hence

$$S_{1N,i_1 i_2 i_3 i_4} = \mathbb{E} S_{1N,i_1 i_2 i_3 i_4} + O_p\left(\sqrt{\operatorname{var} S_{1N,i_1 i_2 i_3 i_4}}\right) = O_p\left(Nh^4\right),$$

under $h \to 0$ and $Nh^{2d} \to \infty$. Similarly we have results for $S_{2N,i_1 i_2 i_3 i_4}$.

Next, write

$$V_{u;i_1 i_2} = w_u \delta_{u,0-1}^{i_1} \delta_{u,0-1}^{i_2} R_u.$$

Note that

$$
\begin{aligned}
&\mathbb{E} V_{u;i_1 i_2} \\
&= \int h^{-2d} \prod_{i=1}^{d} \left(K\left(\delta_{n0}^i/h\right) K\left(\delta_{n1}^i/h\right)\right) \delta_{n,0-1}^{i_1} \delta_{n,0-1}^{i_2} \times \\
&\quad \sum_{1 \le k,l,m,r \le d} \delta_{u,0-1}^m \left(\delta_{n0}^k \delta_{n0}^l - \delta_{n1}^k \delta_{n1}^l\right) F_{klm} \times \\
&\quad f(p^1 + \delta_{n0}^1, \ldots, p^d + \delta_{n1}^d) d\delta_{n0}^1 \ldots d\delta_{n1}^d \\
&= h^5 \int K\left(s_{u0}^1\right) \cdot \ldots \cdot K\left(s_{u1}^d\right) \left(s_{u0}^{i_1} - s_{u1}^{i_1}\right)\left(s_{u0}^{i_2} - s_{u1}^{i_2}\right) \times \\
&\quad \sum_{1 \le k,l,m,r \le d} \left(s_{u0}^{i_2} - s_{u1}^{i_2}\right)\left(s_{u0}^k s_{u0}^l - s_{u1}^k s_{u1}^l\right) F_{klm} \times \\
&\quad \left(f(p^1, \ldots p^d) + h \sum_{r=1}^{d} \frac{\partial f}{\partial p^r}(p)\left(s_{u0}^r + s_{u1}^r\right) + o(h)\right) ds_{u0}^1 \ldots ds_{u1}^d \\
&= O(h^6),
\end{aligned}
$$

where $F_{klm} = \Gamma_{kl}^r G_{mr}$. Indeed, since the kernel $K$ is symmetric, and the leading terms in the integrant is of fifth power of $s$, thus with some abuse of notation, $\mathbb{E}V_{u;i_1 i_2} = h^5 F \int K(s)s^5 (f + hO(s)) \, ds = O(h^6)$. Similarly

$$\operatorname{var} V_{u;i_1 i_2} = O(h^{10-2d}).$$

The rest of the results for $S_{3N,i_1 i_2 i_3 i_4}$ proceeds analogously to that of $S_{1N,i_1 i_2 i_3 i_4}$ and $S_{2N,i_1 i_2 i_3 i_4}$. $\qquad\square$

*Proposition* E.2. Under the conditions of Proposition E.1,

$$\operatorname{bias}\left(\hat{\boldsymbol{\beta}}|\mathbf{X}\right) = O_p\left(h^2\right), \quad \operatorname{var}\left(\hat{\boldsymbol{\beta}}|\mathbf{X}\right) = O_p\left(\frac{1}{Nh^{4+2d}}\right),$$

as $h \to 0$ and $Nh^{2+2d} \to \infty$, where $\mathbf{X}$ are the observed endpoints.

*Proof.* Note that $S_{1N;i_1 i_2 i_3 i_4}$ are elements of $\mathbf{D}^T \mathbf{W} \mathbf{D}$, where one pair of $(i_1, i_2)$ index a row while one pair of $(i_3, i_4)$ index a column for $i_1, i_2, i_3, i_4 = 1, \ldots, d$. Similarly $S_{2N;i_1 i_2 i_3 i_4}$ are elements of $\mathbf{D}^T \mathbf{\Sigma} \mathbf{D}$, and $S_{3N;i_1 i_2}$ are elements of $\mathbf{D}^T \mathbf{W} r$ by Proposition 3.1. Applying Proposition E.1 leads to the result. $\qquad\square$

