# OpenReview forum: "Estimating Riemannian Metric with Noise-Contaminated Intrinsic Distance"
_ICLR.cc/2023/Conference — Submitted to ICLR 2023_

### Official Review · Reviewer_KGPf · 2022-10-24

**Confidence:** 3
**Correctness:** 3
**Technical Novelty And Significance:** 3
**Empirical Novelty And Significance:** 2
**Recommendation:** 3

**Clarity, Quality, Novelty And Reproducibility:**

 [Quality and novelty of the proposed model, Score: 65/100]

The proposed local linear regression model is new and has a solid differential geometry foundation given in Proposition 3.1.

(1). Although Proposition 3.1 seems a combination of lemmas similar to the ones in (de Carmo, 1992) and (Lang, 1999), the expression of Taylor expansion still has its novelty.

(2). It is interesting to find that although (4.1) does not involve the Christoffel symbols, they do affect the performance of estimation. Could the authors give some intuitive explanations?

[Quality regarding empirical use in modern CV&ML applications, Score: 20/100]

In my mind, the underlying assumption of the proposed model is a bit strict and may seriously limit its real applications. The lack of experiments on modern CV&ML datasets may also reflect this disadvantage.

[Clarity and Presentation, Score: 50/100]
This paper is in general clearly written. The somewhat complicated geometric notions are relatively well introduced. However, this paper has typos and not introduced symbols examples as follows:

(1) In Eq. (3.7), the iterator in the summation symbol should be $u$.

(2) In Proposition C.1, the Riemannian curvature tensor $R(v,w)w$ should be defined/explained in advance.

(3) In Line 3, P17, it should be $ H(s)=dist(\gamma_0(s),\gamma_1(s))^2=\|\|T\|\|^2\big|_{s,t}$.

(4) In Line 5, P18, $Rm(\cdot,\cdot,\cdot,\cdot)$ should be introduced.

[Reproductivity, Score: 50/100]

Although no code is uploaded as supplemental materials, the details are relatively sufficient to reproduce the results.

**Strength And Weaknesses:**

[Strength]

1). This paper proposes an interesting linear regression model which stems from a rigorous expression of Taylor approximation to the squared geodesic distances on a Riemannian manifold.

2). Bounds on the bias and variance of the estimator are quantified in an asymptotic sense.

[Weakness]

1). Assumptions behind the local linear model may be too strict for real applications.

The proposed local regression model assumes the samples lie in small neighborhood of another point. However, in real applications using binary similarity measurements in computer vision, perception tasks, and recommendation systems, the compared samples may be far from each other and the assumption of localness may be violated.

2). Insufficient empirical verifications for real applications.

Through experiments on unit spheres, spirals, and taxi duration data, the authors provide nice visualizations of the estimate geometric parameters. However, neither comparison results with other methods nor experimental results on modern CV&ML datasets are given.


**Summary Of The Paper:**

To measure the difference between two samples $A$ and $B$,  this paper assumes the samples $A$ and $B$ are endpoints of two different geodesics starting from the same point $P$ in a small neighborhood and approximates their squared geodesic distances using Taylor extension which leads to a local linear regression model. Geometric characteristics like Riemannian metric and the Christoffel symbols serve as parameters of the proposed linear regression model. For the special case using squared loss, the authors established asymptotic bounds on the bias and variance of the estimator. Some intuitive results are shown through numerical experiments on unit spheres, spirals, and taxi trip duration.

**Summary Of The Review:**

I recommend "reject, not good enough" due to the scores regarding Clarity, Quality, Novelty And Reproducibility.
[Quality and novelty of the proposed model, Score: 65/100]  [Quality regarding empirical use in modern CV&ML applications, Score: 20/100]
[Clarity and Presentation, Score: 50/100] [Reproductivity, Score: 50/100]

---

> ### Author Response · Authors · 2022-11-19
> **Response to Review by Reviewer KGPf (part 1)**
>
> We truly appreciate your careful read of our manuscript and helpful comments. Substantial changes have been made in the manuscript accordingly. The major revision includes:
>
> 1. We have added an example using MNIST (section 7) to illustrate how to utilize the proposal for high-dimensional data (e.g., images), especially when the data are intrinsically low-dimensional.
> 2. We have added a section (section B) to the supplementary materials to clarity several points regarding implementation as prompted by the reviewers. It includes how we select the target points p, how we construct estimation for larger domain via post-smoothing, and how we compute geodesic curves.
> 3. We have corrected typos as we are aware of. We also shortened some sentences and moved some details to the appendix to make space for the added MNIST example.
>
> We summarized your comments and provided a point-by-point response in **part 2** of our response.

---

> > ### Author Response · Authors · 2022-11-19
> > **Response to Review by Reviewer KGPf (part 2)**
> >
> > We summarized your comments and provided a point-by-point response in below.
> >
> > ---
> >
> > - *Assumptions behind the local linear model may be too strict for real application, the compared samples may be far from each other and the assumption of localness may be violated.*
> >
> > Sufficiently dense data is necessary for the proposal, which could be demanding for high-dimensional data due to the curse of dimensionality. However, data often exhibit the manifold phenomenon, namely data intrinsically lie close to a low-dimensional manifold. Thus, we can first embed the data to a low-dimensional space via dimension reduction techniques such as tSNE, and then apply the proposal to the embedded coordinates. The embedded representations tend to be dense since they lie in a low-dimensional space. Thus, dimensional reduction can substantially alleviate the curse of dimensionality, and the dense local neighborhood requirement will more likely hold true.
> >
> > An additional example using MNIST is added to illustrate this point. The results reveal that our estimated geodesics represents shortest paths between images where on the geodesic, images change smoothly with fewer switches in the underlying digit and smaller Wasserstein distance.
> >
> > - *Insufficient empirical verifications for real applications and lack of results on modern CV&ML datasets.*
> >
> > As previously mentioned, we included a new example using MNIST for illustration. In this example, we compare the estimated geodesic with the geodesic given by the Wasserstein transportion. A highlight of the result (Figure 7.1) is that the images along our estimated geodesics are recognizable because the geodesics reside in the space of digits, while the Wasserstein geodesic could produce unrecognizable intermediate states.
> >
> > Our work emphasizes the recovery and interpretation of the intrinsic geometry induced by the observed similarity measurements, while many related distance metric learning works focus more on utilizing the trained distance metric for downstream tasks such as classification (e.g., via kNN). The authors are aware of relatively few CV&ML works that also target the recovery of underlying geometry, either with limited scope (e.g., based on parametric families: Lebanon 2002, 2006; Le & Cuturi 2015), or constructs Riemannian metric by pooling multiple local Mahalanobis distance (e.g., Hauberg et al. 2012). Thus, we did not find other methods directly comparable to ours.
> >
> > - *Although Proposition 3.1 seems a combination of lemmas similar to the ones in (de Carmo, 1992) and (Lang, 1999), the expression of Taylor expansion still has its novelty.*
> >
> > Thank you for your comment.
> >
> > - *It is interesting to find that although (4.1) does not involve the Christoffel symbols, they do affect the performance of estimation. Could the authors give some intuitive explanations?*
> >
> > In analogy to Taylor expansion, this approximation improves its accuracy as the neighborhood gets smaller, even though it neglects the Christoffel symbol term. Adding higher order terms to the expansion can improve approximation, but Equation (4.1) uses a shorter expansion for easier derivation.
> >
> > - *In my mind, the underlying assumption of the proposed model is a bit strict and may seriously limit its real applications. The lack of experiments on modern CV&ML datasets may also reflect this disadvantage.*
> >
> > In sum, the assumptions we made are that
> >
> > 1. The data are drawn from some manifold with low intrinsic dimensions.
> > 2. The observed similarity measures are generated based on intrinsic geodesic distance, as in eq (3.1).
> >
> > The low-dimensional manifold assumption is commonly satisfied by real-world image and audio datasets because of the manifold phenomenon. This enables us to utilize nonlinear dimension reduction so as to alleviate the curse of dimensionality.
> >
> > We would like to clarify that the mathematical backbone of the proposed method is equation (3.1) and (3.5), which are supported by solid theoretical results under mild assumptions. We propose a general framework linking the Taylor expansion (3.5) and the similarity measures via (3.1). The specific models enlisted in Example 3.1 and Example 3.2 are given as commonly encountered scenarios that can be handled by our framework;  the proposed method is not limited to these models. Using different link function g and loss function Q can flexibly accommodate a variety of data generating mechanism similar to what generalized linear model (GLM) can handle.
> >
> > - *This paper has typos and not introduced symbols.*
> >
> > The authors appreciate the careful read by the reviewer. These and other typos have now been corrected. Additional clarifications are added for the Riemannian curvature tensor.

---

> > > ### Comment · Reviewer_KGPf · 2022-12-07
> > > **Keeping my score "3 reject"**
> > >
> > > Many thanks for the authors' feedback. Although some of my concerns have been clarified, I am still sorry to keep my score "3 reject" due to the scores regarding Clarity, Quality, and Novelty.

---

### Official Review · Reviewer_gAuZ · 2022-10-24

**Confidence:** 4
**Correctness:** 4
**Technical Novelty And Significance:** 3
**Empirical Novelty And Significance:** 2
**Recommendation:** 3

**Clarity, Quality, Novelty And Reproducibility:**

I found the paper to be quite clear to read, and the idea seems both sensible and novel. I would be unable to reproduce the results as it is not clear how to compute geodesics.

**Strength And Weaknesses:**

## Strengths
* The derived Taylor expansion is $\mathcal{O}(t^4)$ indicating that it is quite accurate as an estimate of a local metric.
* The method seems both quite general and novel.

## Weaknesses
* I was unable to determine if the estimated Riemannian metric is always positive definite. It would be nice with a clarification on that.
* As far as I can tell, Eq. 3.10 relies on Euclidean distances through the kernel functions. The main motivation of the work is that we should not trust the Euclidean distance, so if my reading is correct, then this seems a bit problematic.
* I was unable to determine how geodesics were computed in practice.
* The paper contains some simple toy experiments to verify that the method works and an application to travel times in Manhattan. While these experiments are nice, I miss a compelling application of the method. I believe that this line of work can be quite useful, but it would be good to demonstrate this to a greater degree.
* I miss a more careful investigation in the synthetic experiments. For example, in the sphere example, it should be trivial to vary the sphere dimensionality and ask, "does the method keep working?" Given that only very low-dimensional problems are considered, I worry that the method does not scale. It would be nice with some evidence to counter such a belief.
* The discussion section of the paper feels very rushed. Perhaps this is the usual time pressure of conference deadlines.

**Summary Of The Paper:**

The paper is concerned with learning a Riemannian metric such that geodesic distances matched (direct or indirect) measured distances. A general framework is presented that captures the common situations of (1) observations are distances, (2) observations are binary indicating if two points are similar or not, or (3) observations are triplets where one pair is more similar than others. It is then proposed to estimate the Riemannian metric through a local Taylor expansion. This is demonstrated to work well on fairly simple low-dimensional problems.

**Summary Of The Review:**

The paper presents an interesting idea, but I find that the experiments are lacking.

---

> ### Author Response · Authors · 2022-11-19
> **Response to Review by Reviewer gAuZ (part 1)**
>
> We truly appreciate your careful read of our manuscript and helpful comments. Substantial changes have been made in the manuscript accordingly. The major revision includes:
>
> 1. We have added an example using MNIST (section 7) to illustrate how to utilize the proposal for high-dimensional data (e.g., images), especially when the data are intrinsically low-dimensional.
> 2. We have added a section (section B) to the supplementary materials to clarity several points regarding implementation as prompted by the reviewers. It includes how we select the target points p, how we construct estimation for larger domain via post-smoothing, and how we compute geodesic curves.
> 3. We have corrected typos as we are aware of. We also shortened some sentences and moved some details to the appendix to make space for the added MNIST example.
>
> We summarized your comments and provided a point-by-point response in **part 2** of our response.

---

> > ### Author Response · Authors · 2022-11-19
> > **Response to Review by Reviewer gAuZ (part 2)**
> >
> > We summarized your comments and provided a point-by-point response in below.
> >
> > ---
> >
> > - *I was unable to determine if the estimated Riemannian metric is always positive definite. It would be nice with a clarification on that.*
> >
> > While the estimated Riemannian metric is not intrinsically guaranteed to be positive definite, our asymptotic theory shows that the estimated metric will be positive definite given a large enough sample size. In our experiments we did not encounter serious issues due to non-positive definiteness, and we expect this would seldom be an issue given sufficient sample. One could in principle adopt constraint optimization to enforce positive definiteness in the estimation, but for simplicity we chose to apply an unrestricted algorithm that is more computationally efficient.
> >
> > - *The main motivation of the work is that we should not trust the Euclidean distance, but, Eq. 3.10 relies on it through the kernel functions.*
> >
> > What we wanted to convey was that a *global* Euclidean distance is unreliable. A smooth manifold is locally linear in the sense that local neighborhoods are Euclidean spaces but slightly bent, and thus a small neighborhood on the manifold can be linearly approximated. This is analogous to that the local regression is able to learn non-linear trends. The Euclidean distance in the kernel functions are used only to construct a small neighborhood w.r.t. the coordinate representation; geodesic distance to the center of the neighborhood are likely smaller for points with similar coordinates, and vice versa. On this small neighborhood we can well approximate the geodesic distance through coordinates using equation (3.5).
> >
> > - *I was unable to determine how geodesics were computed in practice.*
> >
> > We now describe how geodesic curves are computed in the additional Section B in the Appendix. Basically, we numerically solved ordinary differential equations (geodesic equations) w.r.t. the estimated Christoffel symbols. We would also like to clarify that it is unnecessary to obtain for input the geodesic curves corresponding to the pairwise distances, just the triplet of two points and the distance (a non-negative number) is sufficient.
> >
> > - *I miss a compelling application of the method.*
> >
> > We have added a more extensive example using MNIST digits. This example serves two purposes. First, it illustrates the applicability of our method to high-dimensional data like images that actually lie close to some low-dimensional manifold. Second, we infer the geometry of embedded digits (using tSNE) where the similarity between images is measured by a composite measure that is defined to be the sum of a scaled Wasserstein distance and an indicator for whether two images represents different digits. The results reveal that our estimated geodesics represents shortest paths between images where on the geodesic, images change smoothly with fewer switches in the underlying digit and smaller Wasserstein distance.
> >
> > - *I miss a more careful investigation in the synthetic experiments. For example, in the sphere example, it should be trivial to vary the sphere dimensionality and ask, "does the method keep working?" Given that only very low-dimensional problems are considered, I worry that the method does not scale. It would be nice with some evidence to counter such a belief.*
> >
> > There is an intrinsic difficulty in extending our fully nonparametric approach to higher dimensions due to the curse of dimensionality, where the rate of convergence is slower and the number of parameters to be estimated per target point scales like $O(d^2)$ (or $O(d^3)$ if also estimating the Christoffel symbols). For the unit sphere example (section 5.1), we obtained an average relative Frobenius error of 0.007 for the estimated metric for $d = 2$-dimensional case, while the $d = 3$ case with same number of observations has an average relative Frobenius error of 0.078. It is reasonable to anticipate larger error in higher-dimensional space without increasing sample size. We decided not to include this in our already lengthy manuscript.
> >
> > On the other hand, due to the manifold assumption, data are oftentimes intrinsically low-dimensional, in the sense that data points could be relatively well represented via few coordinates. In such case, the proposed method is combined with dimensional reduction techniques to infer the geometry in the low-dimensional embedded space, as shown in the added MNIST example.
> >
> > - *The discussion section of the paper feels very rushed. Perhaps this is the usual time pressure of conference deadlines.*
> >
> > We have now rewritten the discussion section in response to the reviews.

---

### Official Review · Reviewer_LDTm · 2022-10-24

**Confidence:** 4
**Correctness:** 4
**Technical Novelty And Significance:** 4
**Empirical Novelty And Significance:** 2
**Recommendation:** 8

**Clarity, Quality, Novelty And Reproducibility:**


The paper is well written and defines all the quantities that are used in the development. It would be useful to provide some intuitive explanation behind the two terms on the right in Eqn. 3.5.

**Strength And Weaknesses:**


Strengths:

Seems like a good idea to learn the local geometry of a manifold. In case that additional information (pairwise geodesic distances or labels or relative labels) is available and the points are relatively dense, then it seems reasonable to be able to learn the metric from the data. I don't know if any one has taken this approach in the past, but it seems quite novel to me.

Weakness:
•	How are the base points p (the basepoint for Eqn. 3.5) chosen in practice. The underlying approximation is valid only for a small t, so the base p should be close to the given data points. Does the method work if the given points are sparse?

•	What are the practical situations where the pairwise geodesic distances between points on a manifold are given, and one does not know the Riemannian metric? I understand the taxi example but I can’t think of too many other situations.

•	It seems strange that the nonlinearity of the manifold is being learnt through a linear system of equations.

•	The experiments are limited in scope. There are two simulation experiments – one on a unit sphere and one on spiral curves. The third example uses time of travel by taxis. A unit sphere is a special case where it is easy to learn the geometry through only the first order information (points and tangent vectors). It would be been nice to see an example of learning geometry of a more complicated manifold.


**Summary Of The Paper:**

This paper develops a theory for estimating the Riemannian metric tensor for a given set of observations and some addition information. This additional information includes a (noisy) measure of similarity between the given points in a pairwise fashion. Examples of this information includes the geodesic distance, or a binary response about the similarity of types, or a binary response about relative comparison.

The formulation relies on a formula for the intrinsic distance between corresponding points on two geodesics shot from the same base p.  This formula approximates the said distance using the Euclidean chords, Riemannian metric tensor, and the Christoffel symbols. The first term computes the Riemannian metric between the shooting vectors and the second term accounts for the curvature of the metric. The paper seizes on this linear approximation and sets up a regression problem for estimating these matrices from the given data. In this way, it estimates the metric tensors and the Christoffel symbol at point p.

The paper illustrates this solution on two simulated data (a unit sphere and a points on double spirals) and one real data set (time take by taxis between points in New York City.

**Summary Of The Review:**


This is an interesting paper which solves a very specific problem. Estimating Riemannian metrics in general is much more ambitious and challenging but it assumes the knowledge of (noisy) pairwise geodesic distances which limits the scope and  makes the problem easier.

---

> ### Author Response · Authors · 2022-11-18
> **Response to Review by Reviewer LDTm**
>
> We truly appreciate your careful read of our manuscript and helpful comments. Substantial changes have been made in the manuscript accordingly. The major revision includes:
>
> 1. We have added an example using MNIST (section 7) to illustrate how to utilize the proposal for high-dimensional data (e.g., images), especially when the data are intrinsically low-dimensional.
> 2. We have added a section (section B) to the supplementary materials to clarity several points regarding implementation as prompted by the reviewers. It includes how we select the target points p, how we construct estimation for larger domain via post-smoothing, and how we compute geodesic curves.
> 3. We have corrected typos as we are aware of. We also shortened some sentences and moved some details to the appendix to make space for the added MNIST example.
>
> We summarized your comments and provided a point-by-point response below.
>
> ---
>
> - *How are the base points $p$ (the basepoint for Eqn. 3.5) chosen in practice.*
>
> Indeed, the base point $p$ needs to be placed at a location surrounded by relatively dense observations. In our simulations, the taxi trip example, and the newly added MNIST example, the base point $p$ ranges over a dense grid on the manifold of interest (the estimated metric are then extended to the entire manifold through post-smoothing, see Section B in the Appendix for details). In general, if one would like to compute geodesic curves, $p$ needs to be placed at the endpoints of the geodesic as well as in regions where the geodesic may pass over.
>
> If observations around a given location are sparse, to achieve a stable estimation a larger neighborhood must be applied, trading off bias for smaller variance. However, the sparse scenario is unlikely to occur at least in the scenarios we consider. For example, when analyzing images, the apparent high-dimensional data often lie close to a low-dimensional manifold (see, e.g., Roweis & Saul 2000 and Tenenbaum et al, 2000). This manifold phenomenon implies that data are concentrated in a low-dimensional space where neighborhoods tend to be dense. On the other hand, for the purpose of inferring geometry associated with a pre-specified distance, one can increase the data size by introducing more pairwise distance measures at the cost of increased computation.
>
> - *What are the practical situations where the pairwise geodesic distances between points on a manifold are given, and one does not know the Riemannian metric?*
>
> As prompted by you and other reviewers, we included in the revised version a new illustration investigating image data in MNIST. We are interested in how digits gradually deform into each other. Here, the distance measure is easier to specify than the Riemannian metric. Namely, we applied a summed measure of Wasserstein distance and an indicator for whether the two digits are different. The proposed method makes it possible to estimate the Riemannian metric and the geodesics (which is never known in practice!) and thus explore the underlying geometry induced the specified distance measure.
>
> - *It seems strange that the nonlinearity of the manifold is being learnt through a linear system of equations.*
>
> A smooth manifold is locally linear in the sense that local neighborhoods are Euclidean spaces but slightly bent, and thus a *small* neighborhood on the manifold can be linearly approximated. This is analogous to that the local regression is able to learn non-linear trends and that nonlinear dimension reduction methods often approximate a surface locally by a tangent space.
>
> - *The experiments are limited in scope. It would be been nice to see an example of learning geometry of a more complicated manifold.*
>
> We have added another example using MNIST data to further illustrate the proposal on a more complicated manifold. In this example the proposed method infer the geometry using only pairwise distances without knowing even the form of the geodesic curves (or the shortest path along which the distance is evaluated).
>
> - *It would be useful to provide some intuitive explanation behind the two terms on the right in Eqn. 3.5.*
>
> To the RHS of Eqn. 3.5, the first term is the quadratic form computing norm of tangent vectors w.r.t. the Riemannian metric under normal coordinates. The second term is the result of coordinate representation of geodesic curve (see Proposition D.2 and the proof of Proposition 3.1 in Section D of the Appendix); it vanishes only under the normal coordinate where the Christoffel symbols are zero. Without knowing the metric, it is hard to construct normal coordinate, which requests orthogonal basis. In short, this term is a result of coordinate representation.

---

### Decision · Program_Chairs · 2023-01-20

**Decision:**

Reject

**Justification For Why Not Higher Score:**

Experimental section is too weak. Method does not scale.

**Justification For Why Not Lower Score:**

na

**Metareview: Summary, Strengths And Weaknesses:**

The paper proposes a method to recover the parameters of a Riemannian metric using noisy observations of distances between points on the manifold (described through coordinates). While interesting and (re)introducing the ICLR audience to a problem area that was heavily studied about 10 years ago, the paper suffers from a few fatal flaws that do not make it a suitable addition to the program as it is. All experiments are toy, and the authors have no strategy to make this work generalize to datatypes that are of interest. As a result the paper is rejected, but this round of reviews has provided the authors with a lot of feedback.